# Tracking down the molecular architecture of the synaptonemal complex by expansion microscopy

Fabian U. Zwettler[1,3], Marie-Christin Spindler [2,3], Sebastian Reinhard[1], Teresa Klein[1], Andreas Kurz[1], Ricardo Benavente[2✉] & Markus Sauer [1✉]

The synaptonemal complex (SC) is a meiosis-specific nuclear multiprotein complex that is essential for proper synapsis, recombination and segregation of homologous chromosomes. We combined structured illumination microscopy (SIM) with different expansion microscopy (ExM) protocols including U-ExM, proExM, and magnified analysis of the proteome (MAP) to investigate the molecular organization of the SC. Comparison with structural data obtained by single-molecule localization microscopy of unexpanded SCs allowed us to investigate ultrastructure preservation of expanded SCs. For image analysis, we developed an automatic image processing software that enabled unbiased comparison of structural properties pre- and post-expansion. Here, MAP-SIM provided the best results and enabled reliable three-color super-resolution microscopy of the SCs of a whole set of chromosomes in a spermatocyte with 20–30 nm spatial resolution. Our data demonstrate that post-expansion labeling by MAP-SIM improves immunolabeling efficiency and allowed us thus to unravel previously hidden details of the molecular organization of SCs.

[1] Department of Biotechnology and Biophysics Biocenter, University of Würzburg, Am Hubland 97074 Würzburg, Germany. [2] Department of Cell and Developmental Biology Biocenter, University of Würzburg, Am Hubland 97074 Würzburg, Germany. [3]These authors contributed equally: Fabian U. Zwettler, Marie-Christin Spindler. ✉email: benavente@biozentrum.uni-wuerzburg.de; m.sauer@uni-wuerzburg.de

maging technologies are central platforms that drive fundamental research in virtually all disciplines across the biological and medical sciences. However, the diffraction barrier of classical fluorescence microscopy has hindered obtaining high-resolution information about the molecular architecture of protein assemblies and their interrelations. So far, only electron microscopy (EM) techniques provided a spatial resolution that enables the investigation of the molecular composition and structure of multiprotein complexes[1]. Super-resolution microscopy methods can now provide spatial resolution that is well below the diffraction limit of light microscopy approaching virtually molecular resolution[2,3]. Physical expansion of the cellular structure of interest represents an alternative approach to bypass the diffraction limit and enables super-resolution imaging on standard fluorescence microscopes. For this purpose, expansion microscopy (ExM) has been developed and successfully applied to visualize cellular structures with ~70 nm lateral resolution by confocal laser scanning microscopy[4].

The original ExM protocol used functionalized antibody–oligonucleotide conjugates that bind to target proteins and cross-link covalently into a swellable hydrogel during polymerization. After degradation of native proteins by enzymatic proteolysis, the sample expands ~4.5-fold in water[4]. To circumvent fluorophore loss during polymerization and protease digestion alternative ExM protocols have been introduced enabling imaging of proteins, RNA, and bacteria in cultured cells, neurons, and tissues also in combination with super-resolution microscopy[5–11]. For example, protein-retention ExM (ProExM)[6] and magnified analysis of the proteome (MAP)[7] have been developed that cross-link proteins themselves into the polymer matrix. Replacing protein digestion by heat and chemical induced denaturation allows post-expansion immunolabeling of chemically embedded proteins. To further increase the achievable resolution, the expansion factor has been increased up to 20-fold[12,13]. However, such high expansion factors dramatically reduce the labeling density and consequently also the achievable structural resolution and require ultimately single-molecule sensitive imaging methods to visualize such extremely diluted fluorescence signals. Furthermore, some doubts remained concerning uniform three-dimensional (3D) expansion and preservation of ultrastructural details especially of multiprotein complexes. Very recently, it has been shown that various expansion protocols do not completely preserve the 3D molecular architecture of centrioles. Only by careful optimization of the expansion protocol ultrastructural details of centrioles could be truthfully preserved by U-ExM[14].

Recently, ExM in combination with structured illumination microscopy (SIM) has been used to investigate the three-dimensional (3D) organization of *Drosophila* synaptonemal complexes (SCs) with a lateral resolution of ~30 nm[9,15]. To locate the expanded sample as close as possible above the coverslip, they had to be dehydrated, cryosectioned into 10 μm sections, and then again expanded and mounted on a coverslip. Finally, SIM with an oil-immersion objective and minimal spherical aberration has been performed. Furthermore, ExM has been combined with 2D single-molecule localization microscopy to elucidate the molecular organization of the murine chromosome axis of SCs on nuclear spreadings with 10–20 nm lateral resolution[16]. However, with a ~3–4x expansion factor in combination with a 2-fold increase in spatial resolution provided by SIM, currently Ex-SIM represents the method of choice for 3D multicolor super-resolution imaging of multiprotein complexes such as SCs.

In the present study, we tested the suitability of different ExM protocols for investigation of the molecular architecture of mammalian synaptonemal complexes (SCs) by SIM. With EM[17–21] and single-molecule localization microscopy[22] data available about the distribution of SC proteins, it is ideally suited as a benchmark structure to evaluate isotropic expansion and structure preservation of different ExM protocols. We developed a robust workflow for Ex-SIM on nuclear spreadings together with an automated image-processing software to simplify the implementation of multicolor Ex-SIM and refined data analysis. The developed method allowed us to unravel new details of the molecular organization of SCs.

## Results and discussion

**Analyzing SIM images of expanded SCs**. Synaptonemal complexes (SCs) are meiosis-specific multiprotein complexes that are essential for synapsis, recombination, and segregation of homologous chromosomes, resulting in the generation of genetically diverse haploid gametes, the prerequisite for sexual reproduction[23,24]. The SC exhibits an evolutionarily conserved ladder-like organization composed of two lateral elements (to which the chromatin of homologous chromosomes is associated) and a central region. In mouse, the central region is formed by a central element running between the lateral elements, and numerous transverse filaments connecting the lateral elements and the central element (Fig. 1). Early EM 3D reconstructions show the ribbon-like lateral elements (LEs) of the SC spanning across the nucleus while turning around the own axis[17–20]. In mammals, eight SC protein components have been identified so far: the proteins SYCP2 and SYCP3 of the lateral elements, SYCP1 of transverse filaments and the proteins SYCE1, SYCE2, SYCE3, TEX12, and SIX6OS1 of the central element[23,24]. The assembly of the SC proteins into an elaborate molecular architecture is hereby tightly coordinated with essential meiotic processes and therefore conserved across species[23,24]. Consequently, localization maps of SC proteins are required to unravel the function of the molecular architecture of the SC in synapsis and recombination and thereby the overall success of meiosis.

In order to elucidate the precise molecular architecture of the SC, nanoscale resolution provided by either EM or super-resolution microscopy is required. Using immunolabeling and super-resolution microscopy by *d*STORM, the position of different proteins of the SC on nuclear spreadings have been visualized with ~20–30 nm lateral resolution. The images revealed that the lateral element protein SYCP3 shows a bimodal distribution separated by $221.6 \pm 6.1$ nm (SD)[22]. This value is in accordance with distances measured between the centers of the two ribbons of parallel oriented lateral elements by EM[17–20]. With SC protein distribution data available, structure preservation and uniform expansion of different ExM protocols can now be efficiently evaluated.

To avoid dehydration and cryosectioning steps and enable super-resolution imaging of expanded SCs by SIM, we first optimized the sample handling. Therefore, we used nuclear spreadings of mouse spermatocytes, a widely used technique to study nuclear proteins, specifically in the field of meiosis. The spreading of the SCs directly onto the surface of the coverslip results in the localization of the proteins close to the coverslip post-expansion and thus avoids dehydration and cryosectioning of the sample enabling the use of high numerical aperture objectives for imaging of hydrogels without spherical aberrations. Further, we adapted the hydrogel composition that enabled us to completely detach and transfer the entire sample from the glass surface to the hydrogel.

We started with immunolabeling of SYCP3, the N termini of SYCP1 (SYCP1N), and SYCE3 as proteins of the lateral element, the transverse filaments, and the central element of the SC, respectively, on nuclear spreadings using three different ExM protocols (Fig. 1). To automatically and objectively analyze the

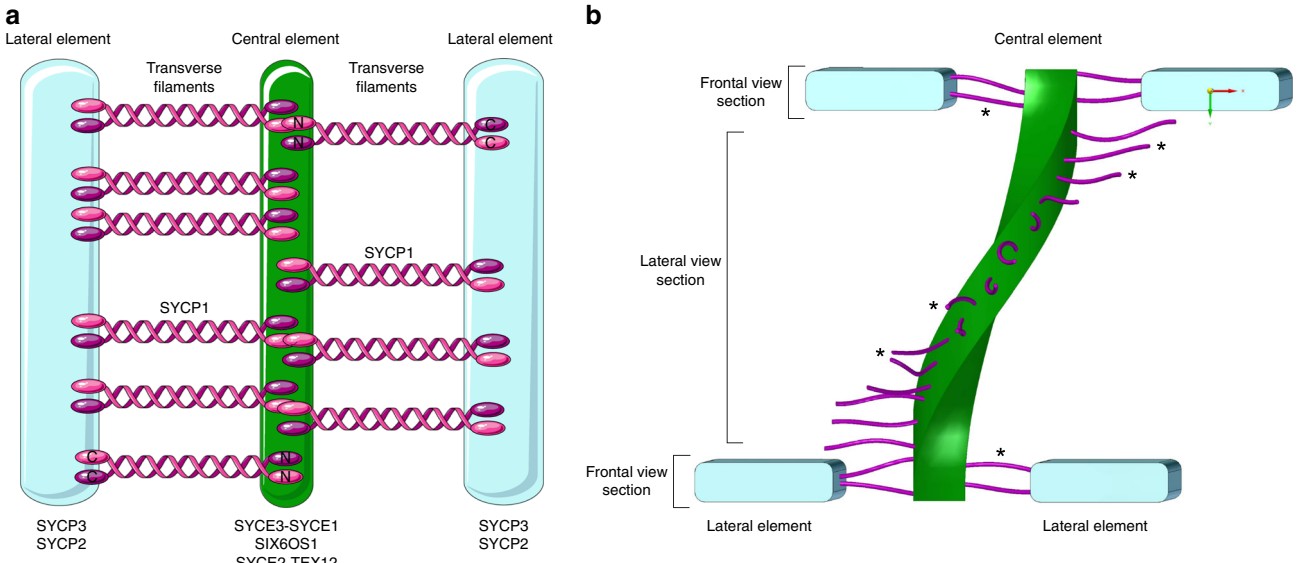

**Fig. 1 The murine synaptonemal complex. a** Schematic representation of the tripartite SC structure of mouse showing the two lateral elements (LEs) consisting of SYCP2 and SYCP3 flanking the central element composed of SYCE1/2/3, Tex12, and SIX6OS1. The transverse filament protein SYCP1 is connecting the lateral element and the central element with the SYCP1 C terminus residing in the lateral and the N terminus in the central element[32]. **b** Schematic representation of the helical structure of the SC exposing frontal view sections and lateral view sections. Lateral elements (SYCP3, SYCP2) in blue, central element (SYCE3-SYCE1, SIX6OS1, and SYCE2-TEX12) in green, transverse filaments (SYCP1) in purple in accordance with **a**. Individual transverse filaments are further marked with asterisks. The lateral elements are not displayed in the lateral view section to expose the central element and transverse filaments.

average position of fluorescently labeled SC proteins and determine distances between bimodal distributed proteins from cross-sectional profiles, we developed 'Line Profiler', an automated image-processing software[25]. Line Profiler uses several image-processing algorithms to evaluate potential regions of interest in 2D maximum intensity projected images. In a first step, all structures in the SYCE3 channel are reduced to lines with one pixel width by using a threshold and a skeletonize algorithm. The resulting pixel coordinates are sorted and fitted with a c-spline. This gives an analytical description of the orientation of SYCE3 and therefore a good approximation for the center (line coordinates) and overall orientation of the helically arranged SYCP3 protein (2-channel mode) (Supplementary Fig. 1). Note that it is also possible to determine the orientation using solely the SYCP3 channel, if the SYCE3 channel cannot be evaluated (1-channel mode) (Supplementary Fig. 2).

To compute the distance between the SC strands, we applied a floodfill algorithm to the SYCP3 channel, leaving only areas embedded in closed shapes unequal to zero. This allowed us to identify regions, where the helical structure of the SC is in plane (regions of maximal distance between the strands). All line coordinates outside of these areas are discarded. A line profile is subsequently constructed for each remaining line coordinate perpendicular to the derivative of the c-spline. For averaging, the line profiles are post aligned to the center between their peaks. As Line Profiler uses 2D images, it is biased to underestimate the true peak-to-peak distance due to profiles taken close to the helical twist. Therefore, we histogrammed the distances of all profiles and determined a more precise value by fitting the curves to the upmost peak (see Methods for more details). We compared the distances between the maximum intensities of the two SYCP3 strands of unexpanded and expanded SCs imaged by *d*STORM and SIM for different expansion protocols, respectively (Fig. 2). To evaluate the expansion factor, we used the 1-channel mode because SYCP3 was the only label used in the reference *d*STORM experiments of unexpanded SCs. SYCP3 bimodal

protein distributions of unexpanded SCs imaged with *d*STORM resulted in an average strand distance of 220.0 ± 21.6 nm (SD) consistent with our previous *d*STORM data[22] (Supplementary Fig. 3).

**Optimization of SC expansion.** We tested the pre-expansion labeling protocol proExM[6], and the two post-expansion labeling protocols MAP[7] and U-ExM[14]. Among the three ExM protocols tested, MAP outperformed the other protocols resulting in an average peak-to-peak distance of cross-sectional profiles of the SYCP3 signals of 650.0 ± 50.0 nm (SD) determined using the 1-channel mode to analyze the cross-sectional profiles (Supplementary Fig. 2). With a bimodal distribution separated by 220.0 nm ± 21.6 nm (SD) as measured by *d*STORM from unexpanded SCs (Supplementary Fig. 3), the MAP protocol showed a ~3.0× increase in the measured SYCP3 peak-to-peak distances. U-ExM enabled post-expansion labeling with various fluorophores and three-color SIM but expanded SCs showed structural breaks indicating insufficient incorporation of proteins into the gel matrix (Supplementary Fig. 4). Average peak-to-peak distances of cross-sectional profiles of the SYCP3 signals of 540 ± 31.3 nm (SD) showed a ~2.5× increase in peak-to-peak distance using the 1-channel mode method (Supplementary Fig. 5). On the other hand, proExM provided the largest increase in peak-to-peak distance of ~4.0× determined by using the 1-channel mode (Supplementary Fig. 5)[4,6].

To verify isotropic expansion, we used the MAP protocol and investigated pre- and post-expansion images of the same spermatocytes immunolabeled for SYCP3 on nuclear spreadings (Supplementary Fig. 6). By comparing pre- and post-expansion images, we determined a macroscopic expansion factor of ~4.2× (Supplementary Fig. 6k). The overlapped images and corresponding transformation matrices showed minimal distortions introduced by the physical expansion process (Supplementary Fig. 7). Furthermore, our MAP-SIM data showed that the molecular

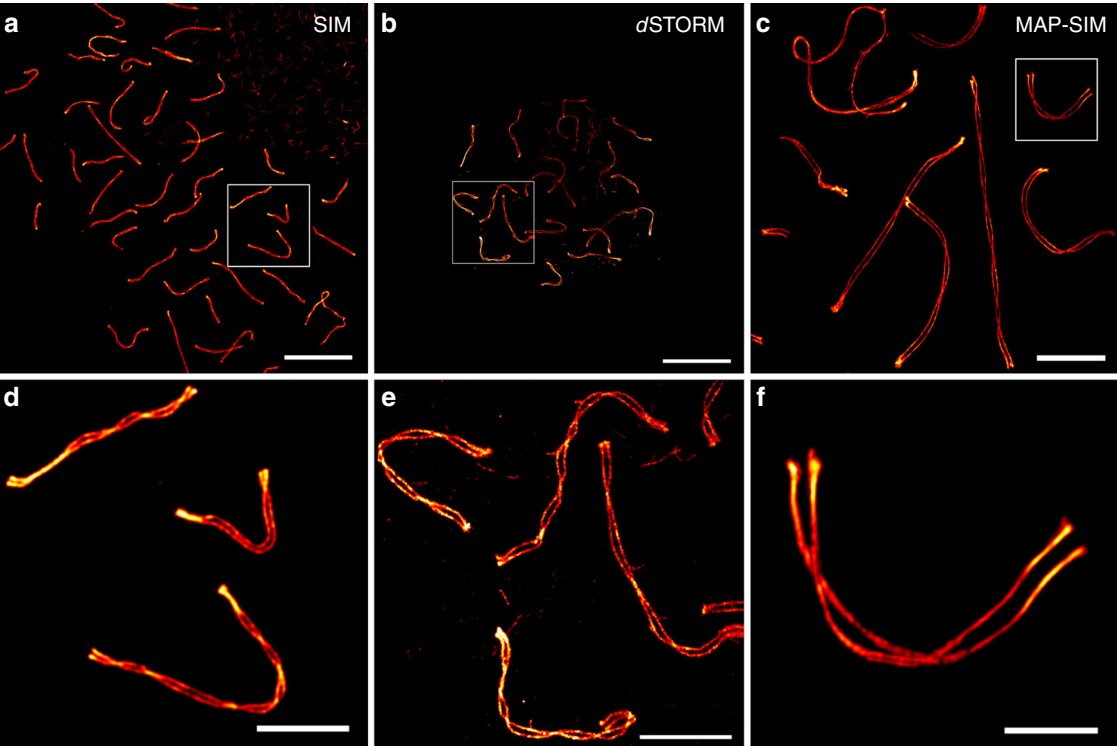

**Fig. 2 Super-resolution imaging of the lateral element protein SYCP3. a** SIM image of unexpanded SYCP3 labeled with Alexa Fluor 568. **b** dSTORM image of unexpanded SYCP3 labeled with Alexa Fluor 647. **c** Expanded MAP-SIM SYCP3 signal (maximum intensity projection) labeled with SeTau647. **d–f** Magnified views of boxed regions in **a–c**, respectively. The images indicate that MAP-SIM provides a similar labeling density and spatial resolution as dSTORM of 20–30 nm in the imaging plane. Scale bars, **a–c** 10 μm, **d–f** 3 μm.

architecture of the SCs is fully preserved demonstrating isotropic expansion (Fig. 3). For example, the SYCP3 signal showed a clear helical organization with the two strands separated by 667.0 nm ± 7.1 nm (SD) (Fig. 3a–c).

The different expansion factors of ~4.2× determined from pre- and post-expansion images of SCs and ~3.0× (Supplementary Fig. 6) resulting from the comparison of peak-to-peak distances of cross-sectional profiles of the lateral element SYCP3 signals of MAP-SIM (Supplementary Fig. 2) and unexpanded dSTORM images (Supplementary Fig. 3), appear confusing but can be well explained by differences in labeling and expansion protocols. The lateral elements are separated by 200 nm, exhibit a width of 37.1 nm and a depth of 100 nm as previously determined by EM experiments in murine spermatocytes (Supplementary Fig. 8)[26,27]. However, the SC is broadened on both sides by immunostaining with primary and secondary IgG antibodies[28]. Therefore, peak-to-peak distances in unexpanded dSTORM images of the lateral element SYCP3 are determined to ~220 nm (Supplementary Fig. 3). We speculate that the SYCP3 epitopes at the periphery of the SC are mainly labeled, while the epitope accessibility is more limited at the center of the SC potentially due to the binding of cohesin complexes at these sites[26,27]. Analysis of the SYCP3 signals measured post-expansion following the proExM protocol[6] revealed a separation distance of ~870 nm (Supplementary Fig. 5e). This increase in distance corresponds to an expected ~4× macroscopic expansion, as the pre-expansion introduced labels expand with the gel expansion factor.

On the other hand, both MAP[7] and U-ExM[14] protocol use post-expansion immunolabeling of SYCP3. We hypothesize that antibodies also reach epitopes in the center of the SC that are inaccessible in the unexpanded state (Supplementary Fig. 8). In addition, the primary and secondary antibodies are not expanded

and thus do not contribute to the broadening of cross-sectional profiles. A further decisive advantage is that post-expansion labeling with primary and secondary antibodies reduces the linkage error dependent on the actual expansion factor to only a few nanometers[29]. Both, the reduced linkage error and improved epitope accessibility of post-expansion MAP result in a shorter peak-to-peak distance determined from cross-sectional profiles of 650 nm (Supplementary Figs. 2 and 8).

Furthermore, the different results obtained by the MAP and U-ExM protocol (Fig. 2 and Supplementary Fig. 4) appear surprising in light of our recent study evaluating the structure preservation of centrioles using different expansion protocols[14]. For centrioles, the MAP protocol was unsuited, as MAP-treated centrioles appeared even smaller when compared to unexpanded samples. Therefore, U-ExM has been introduced as a variation of the MAP protocol using weaker fixation that enabled excellent isotropic expansion and preservation of the centriole ultrastructure[14]. The main difference between centrioles and SCs is their biomolecular composition. Although centrioles are solely composed of proteins, the SC consists of a tight association between DNA and proteins, which may affect expansion efficiency and isotropy. Using the MAP protocol, proteins are denatured after gelation at high temperatures using sodium dodecyl sulfate (SDS). Apart from denaturation, SDS also confers its negative charge to proteins[30,31]. As the DNA is negatively charged as well, crosslinked DNA and proteins repel each other. We hypothesize that the repulsion between DNA and proteins facilitates isotropic expansion of SCs using the MAP protocol.

Overall, MAP-SIM with an optimized gel composition enables isotropic ~4.2× expansion, efficient transfer of proteins into the hydrogel and molecular structure preservation of SCs. In combination with a twofold resolution enhancement of SIM,

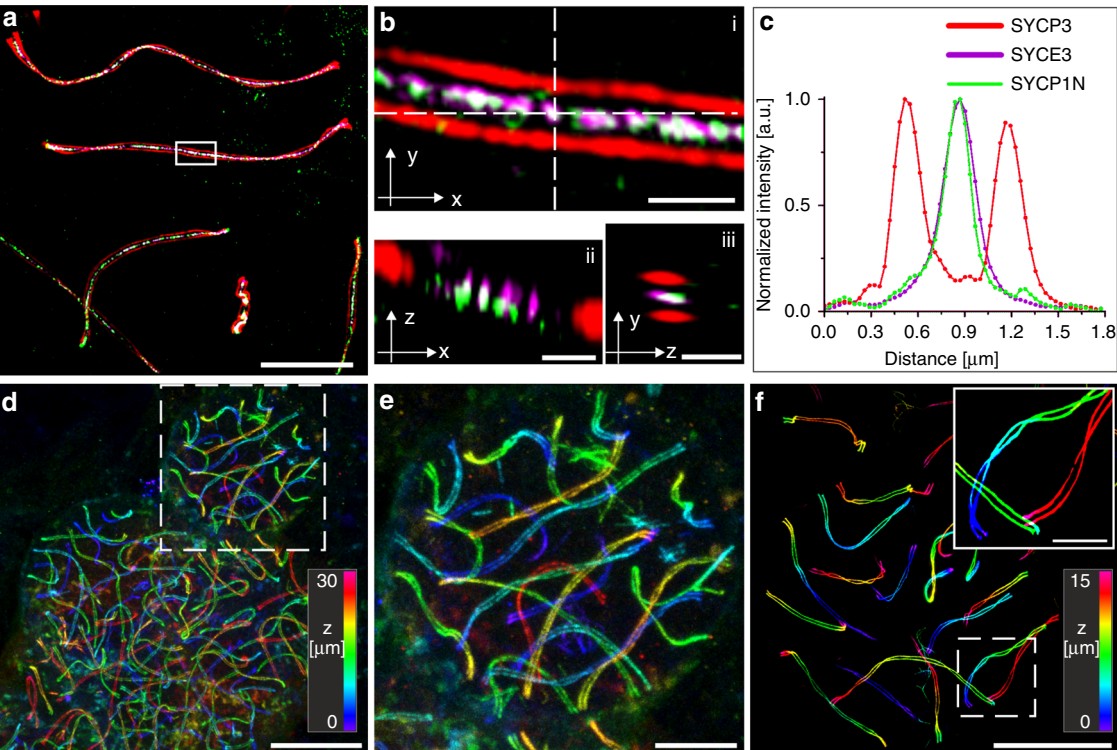

**Fig. 3 3D-multicolor MAP-SIM of SYCP3, SYCP1 N terminus, and SYCE3. a** SIM image of post-expansion SeTau647 labeled SYCP3 as a component of the lateral element (red), the transverse filament SYCP1 N terminus labeled with Alexa Fluor 488 (green), and SYCE3 of the central element labeled with Alexa Fluor 568 (magenta). **b** Magnified views of white boxed region in (**a**). ii Orthogonal view of horizontal white dashed line in i. iii Orthogonal view of vertical white dashed line in (**i**). **c** Transversal intensity profile perpendicular to the orientation of the SC shown in (**b**). The selected section exhibits a bimodal distribution of the SYCP3 signal separated by 667.0 nm ± 7.1 nm (SD). The SYCP1 N terminus and SYCE3 signals of the section (**b**) show monomodal distributions with FWHM of 214.7 ± 6.9 nm (SD) and 258.3 ± 4.8 nm (SD), respectively. **d** Large field of view (100 × 100 × 30 μm[3]) 3D-re-scan confocal microscopy image of spreadings. SYCP3 was labeled post-expansion with SeTau647. **e** Magnified view of boxed region in (**d**). (**f**) 3D-MAP-SIM image of the SCs of an entire set of chromosomes in a spermatocyte visualized by post-expansion labeling of SYCP3 with SeTau647. The inlet shows the enlarged view of the boxed region in (**f**). Scale bars, (**a**) 10 μm, (**b**), i–iii 1 μm, (**d**) 25 μm, (**e**) 10 μm, (**f**) 15 μm, (**f**), inlet 5 μm.

MAP-SIM provides a similar spatial resolution as *d*STORM[22] but in addition a higher immunolabeling efficiency because of the improved epitope accessibility of post-expansion protocols and a smaller linkage error[14,29]. Therefore, we used our optimized MAP-SIM approach (Supplementary Fig. 9) in all following experiments to investigate details of the molecular architecture of SCs.

**3D 3-color MAP-SIM imaging of SCs.** Previous *d*STORM and immunogold EM experiments have shown that the C terminus of SYCP1 localizes to the inner edge of the lateral element and the N termini of SYCP1 interact in the central element[22,32]. These findings are in accordance with recent STORM experiments performed on 2.7× expanded nuclear spreadings[16]. Here, the SYCP1 N terminus was located in the central element roughly 110 nm away from the SYCP3 labeled lateral element (LE), whereas the SYCP1 C terminus localized 25 nm more inward to SYCP3, which corresponds to the inner edge of the lateral element[16]. Further, in *d*STORM experiments the width of the monomodal localization distributions of the N terminus of transverse filament protein SYCP1 and the central element protein SYCE3 were determined to 39.8 ± 1.1 nm (SD) and 67.8 ± 2.1 nm (SD), respectively, in frontal views of the SC[22]. The broader signal distribution of SYCE3 localizations indicates that the interaction of SYCP1 and SYCE3 might not be limited to the N terminus of SYCP1. This is consistent with expanded MAP-SIM

protein distributions of 160.2 ± 1.1 nm (SD) for SYCP1N and 207.5 ± 0.78 nm (SD) for SYCE3 measured in frontal view sections of the SC (Supplementary Fig. 2e).

As *d*STORM requires efficient photoswitching of organic dyes in oxygen-depleted thiol-buffer, it is currently limited to two-color experiments, whereby carbocyanine dyes such as Cy5 and Alexa Fluor 647 are the best suited fluorophores[33,34]. In contrast, MAP-SIM provides similar resolution but enables imaging with up to three colors simultaneously without optimization of the photoswitching buffer conditions. Thus, only the number of available laser lines limits multicolor super-resolution microscopy experiments. Therefore, we immunolabeled the same structural features of the SC (the N terminus of SYCP1, SYCE3, and SYCP3) as in previous *d*STORM experiments also in triple-localization MAP-SIM experiment (Fig. 3a–c). The MAP-SIM images clearly exhibited similar details of the molecular architecture of SCs as single-molecule localization microscopy of unexpanded samples. In addition, the images confirmed isotropic expansion and preservation of the molecular structure of the SC. Intriguingly, spreading of SCs in combination with MAP allowed us to acquire 3D super-resolution images of large expanded samples, e.g. 100 × 100 × 30 μm[3] (200 nm z-steps) by re-scan confocal microscopy (RCM)[35] (Fig. 3d, e) and 80 × 80 × 15 μm[3] (110 nm z-steps) and larger by SIM, i.e. imaging of the SCs of a whole set of chromosomes in a spermatocyte with detailed structural information of single chromosomes (Fig. 3f).

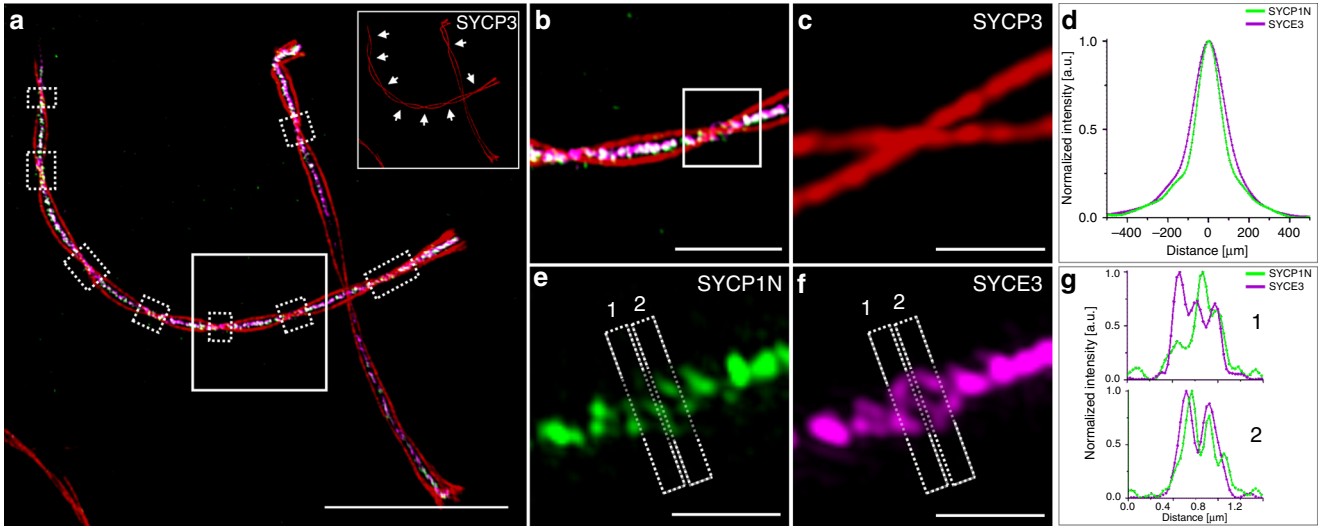

**Fig. 4 MAP-SIM reveals a complex organization of the SC central region. a** MAP-SIM image of SYCE3 labeled with Alexa Fluor 568 (magenta), SYCP1N labeled with Alexa Fluor 488 (green) and SYCP3 labeled with SeTau647 (red). Lateral view sections of the SC are visualized as twists in the SYCP3 signal (marked by white dotted boxes and indicated by arrows in the inlet of (**a**) showing only the SYCP3 channel). **b** Magnified view of white boxed region in (**a**. **c**, **e**, **f**) Enlarged views of highlighted region in (**b**) showing the SYCP3 signal in red (**c**), the SYCP1N signal in green (**e**) and the SYCE3 signal in magenta (**f**) with two sites selected for protein distribution analysis (1, 2). **d** Averaged profile of all cross-sectional intensity profiles along the SYCE3 (magenta) and SYCP1N (green) signals analyzed at regions specified in (**a**) by white dotted boxes showing a monomodal signal distribution of 168.0 ± 1.1 nm (SD) for SYCP1N and 211.43 ± 0.89 nm for SYCE3 derived from single Gaussian fitting. **g** Cross-sectional intensity profiles of lateral view sections 1 and 2 of the SC (indicated in **e** and **f**) show a multimodal organization of the central element protein SYCE3 (magenta) and the N terminus of SYCP1 (green). Scale bars, (**a**) 10 µm, (**b**) 3 µm, (**c**, **e**, **f**) 1 µm.

**MAP-SIM reveals a complex network organization of the SC central region.** MAP-SIM images of lateral views of the SC reveal the topology of the transverse filament protein SYCP1 that is oriented perpendicularly to the lateral element protein SYCP3 and the central element protein SYCE3. dSTORM and immunogold EM images showed a bimodal distribution of the N terminus of SYCP1 and SYCE3 in lateral view sections of the SC[21,22]. EM tomography uncovered a multilayered organization of the central element in insects[36,37]. On the other hand, EM tomography based 3D models of the murine SC recently revealed the absence of a layered organization of the SC central region. Instead, both a visual and mathematical analysis pointed towards an unstructured organization of the central element proteins[27].

Therefore, we next analyzed the signal distribution of the N terminus of the transverse filament protein SYCP1 by MAP-SIM of nuclear spreadings in more detail. In addition to analyzing the distribution of proteins in frontal views, we also generated line profiles of SC proteins from lateral views (Fig. 4a–c, Supplementary Fig. 10, and Supplementary Movie 1). In agreement with EM tomographic data[38], the signal distributions of the MAP-SIM imaged SYCP1 N terminus and SYCE3 indicated a far more complex distribution of the central region proteins than the bimodal distribution previously described by dSTORM and immunogold EM[21,22]. Averaged cross-sectional intensity profiles of SYCE3 and SYCP1N (Fig. 4e, f) clearly showed a monomodal distribution of the two proteins (Fig. 4d and Supplementary Fig. 10) in accordance with the unstructured organization of the transverse filaments into a large single layer as revealed by EM tomography in a previous study[27]. On the other hand, in some lateral view sections of MAP-SIM images of the SYCP3 signal the N terminus of the transverse filament protein SYCP1 and the central element protein SYCE3 revealed bi- and multimodal signal distributions supporting a multilayered organization of the central element (Fig. 4g and Supplementary Fig. 10). The occasional disclosure of multimodal distributions in some lateral view sections of MAP-SIM expanded SCs can be attributed to the high labeling density provided by MAP. Combined with the post-processing super-resolution of SIM MAP-SIM enables to resolve individual SYCP1 positive transverse filaments, which is reflected in the detected multimodal distributions (Supplementary Fig. 10).

The central element is on average 26 by 38 nm and comprises at least seven proteins in the mouse. In traditional immunofluorescence and immuno-EM preparations, antibody–antibody–fluorophore and antibody–antibody–gold complexes localize in a radius of 20–25 nm around the central antigen[38–41]. After binding of the first complexes to their respective epitopes, free epitopes at the core of the central element are likely difficult to access. Using MAP-SIM, epitope accessibility is improved due to the initial physical enlargement of the multiprotein complex before post-expansion labeling. The resulting higher labeling density, specifically at the core of the central element, in combination with the reduced linkage error of post-expansion labeling[29] provides a more fine-grained resolution of the molecular organization of the central element. Hence, MAP-SIM discloses new information about the molecular organization of the SC, particularly that the central element proteins are not organized unambiguously as distinct layers but they form a complex network composed of the transverse filament protein SYCP1 and the central element protein SYCE3 as well as other central element proteins in agreement with recent electron tomographic findings[27].

**Molecular details of SC axes uncovered by MAP-SIM.** One striking morphological feature of the lateral elements, which could not be visualized before by light microscopy, is their occasional splitting into two or more sub lateral elements (subLEs). Variations of subLEs have been observed across species from animals to plants to yeast in various EM preparations (histological sections, whole-mount preparations, spreadings)[17,42–48]. Again, the continuous signal distribution of MAP-SIM allowed us to close the existing gap between light microscopy and EM and resolve the splitting of the two SYCP3 strands in murine pachytene

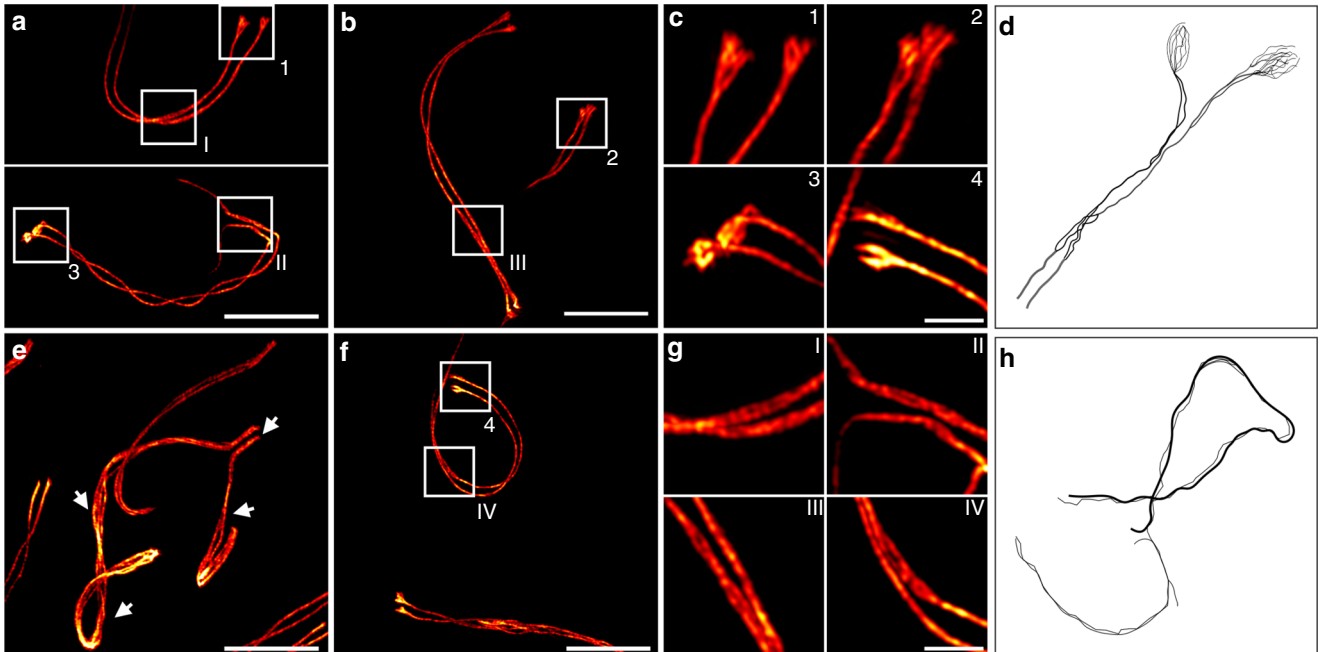

**Fig. 5 Structural details of the SC lateral element revealed by MAP-SIM of SYCP3. a–c, f, g** The SYCP3 signal shows occasional bifurcation along the two SC strands (I–IV) and various degrees of fraying at their ends depending on the respective pachytene stage (1–4). **e** Unpaired regions of the *XY* pair (arrowheads) also show a strong degree of fraying. **d, h** Schematic representations of early EM reports of two or more sub lateral elements (subLEs) in mammals[46,48] in accordance with the splitting of the SYCP3 signal resolved by MAP-SIM in this study. **d** Representation of the lateral element strand splitting in two and fraying at its end according to literature[46,48]. **h** Fraying of LEs associated with unpaired regions of the XY pair modeled according to literature[48]. Scale bars, (**a**, **b**) 7 μm, (**c**) 1.5 μm, (**e–f**) 7 μm, (**g**) 1.5 μm.

spermatocytes (Fig. 5, Supplementary Figs. 11 and 12, and Supplementary Movie 2). Spermatocytes contain 40 chromosomes that synapse as bivalents of homologous chromosomes along the synaptonemal complex in order to recombine. The fully synapsed bivalents hereby span the nuclear space with the two pairs of telomeres residing at distant sides of the nuclear envelope. In the *XY* pair, synapsis is confined to the pseudo-autosomal region and is therefore incomplete. In autosomes, MAP-SIM resolved a doubling of each of the SYCP3 strands (i.e. LEs) specifically, but not exclusively, in lateral view sections of the SC and at the ends of the SYCP3 strands (Fig. 5, Supplementary Figs. 11 and 12 and Supplementary Movie 2).

With progression of prophase I, the degree of fraying at the end of the SYCP3 strands increases. In our experiments, we observed a bifurcation of the SYCP3 ends in mid-pachynema and a fraying of the ends into multiple strands in late pachynema/early diplonema (Fig. 5, Supplementary Figs. 11 and 12 and Supplementary Movie 2). These observations are in agreement with EM findings of subLEs in murine spreadings[45]. Here, a multistranded organization of the lateral elements that arrange into two compact subLEs through interaction with the two sister chromatids has been proposed[44]. In EM images of silver-stained spreadings of mouse SCs, both axes appeared double or multistranded with a higher frequency of subLEs at unpaired regions[48]. Using MAP-SIM, we resolved for the first time the reported fraying of the axes into SYCP3 positive fibrils by light microscopy. In accordance with the conservation of the subLEs of the autosomes, also the fraying of the *XY* axes is common in mammals[49,50].

In mid-pachynema, the homologs are fully synapsed along the SC. Here, doubling of lateral elements is frequently observed and potentially related to the association with the two sister chromatids of the homologs. In diplonema, desynapsis starts

and SCs gradually disassemble. At this stage, the lateral elements appear to disperse into individual SYCP3 fibrils. A strong degree of fraying has also been observed in unpaired regions of the *XY* pair. In 2014, Syrjänen et al. resolved the crystal structure of human SYCP3[51]. They showed that the tetrameric protein is approximately 20 nm long and is organized in an antiparallel arrangement that exposes its N-terminal DNA binding sites at either end. Binding to stretches of DNA, SYCP3 self-assembles into higher-order fibers that resemble the lateral elements in vitro. Based on doubling of the lateral elements observed in EM, it has been speculated that SYCP3 might assemble into one subLE per sister chromatid to prevent recombination between sister chromatids while maintaining chromatin cohesion[51]. The resolution of the splitting of the SC strands is an example for the degree of molecular detail that can be resolved using MAP-SIM overall to uncover new areas of investigation in cell biology. Our data show that through isotropic structure preserving expansion combined with SIM, ultrastructural details of SCs can be revealed with standard immunofluorescence techniques in common sample preparations at axial depths of up to ~60 μm using high numerical aperture water-immersion microscope objectives.

Super-resolution microscopy techniques such as *d*STORM and PALM have enabled new insights into how proteins are organized in cells and multiprotein complexes, with a spatial resolution approaching virtually EM level[2,3]. Nevertheless, structural details of the molecular architecture of multiprotein complexes remained largely accessible only to EM methods. This inability of super-resolution microscopy methods is often due to insufficient structural resolution, which is ultimately controlled by the labeling density. Our results demonstrate that post-expansion labeling protocols reduce the linkage error. Furthermore, the more continuous labeling of SC filaments observed by post-

expansion labeling suggests a higher epitope accessibility for IgG antibodies resulting in improved labeling densities especially of sterically demanding multiprotein complexes[14,29]. Consequently, MAP-SIM provides enhanced ultrastructural resolution as demonstrated here for an important DNA-associated multi-protein complex.

## Methods

**Ethics Statement**. Animal care and experiments were conducted in accordance with the guidelines provided by the German Animal Welfare Act (German Ministry of Agriculture, Health and Economic Cooperation). Animal housing, breeding and experimental protocols were approved by the regulatory agency of the city of Wuerzburg (Reference 821-8760.00-10/90 approved 05.06.1992; according to §11/1 o.1 of the German Animal Welfare Act) and carried out following strict guidelines to ensure careful, consistent, and ethical handling of mice.

**Reagents**. Acrylamide (AA, 40%, A4058, Sigma), Acryloyl-X, SE, 6-((acryloyl) amino)hexanoic Acid, Succinimidyl Ester (A20770, ThermoFisher), Agarose (A9539, Sigma), Ammonium persulfate (APS, A3678, Sigma), Bovine Serum Albumin (BSA, A2153, Sigma), Dimethyl sulfoxide (DMSO, D12345, Thermo-Fisher), Ethanol (absolute, ≥99.8%, 32205, Sigma), Ethylenediaminetetraacetic acid (EDTA, E1644, Sigma), Formaldehyde (FA, 36.5–38%, F8775, Sigma), Guanidine hydrochloride (guanidine HCl, 50933, Sigma), Glucose (G8270, Sigma), Glucose oxidase (Sigma), β-Mercaptoethylamine (MEA, M9768, Sigma), $N,N'$-methylen-bisacrylamide (BIS, 2%, M1533, Sigma), $N,N,N',N'$-Tetramethylethylenediamine (TEMED, T7024, Sigma), PBS (P5493, Sigma), Poly-D-lysine hydropromide (P6407, Sigma), Polyoxyethylene (20) sorbitan monolaurate solution (Tween-20, 10%, 93774, Sigma), Potassium hydroxide (P1767, Sigma), Proteinase K (P4850, Sigma), Sodium acrylate (SA, 97–99%, 408220, Sigma), Sodium chloride (NaCl, S7653, Sigma), Sodium dodecyl sulfate (SDS, L3771, Sigma), Tris base (T6066, Sigma), Triton X-100 Surf-Amps Detergent Solution (10% (w/v), 28313, ThermoFisher), Tween-20 Surf-Amps Detergent Solution (Tween-20, 28320, Sigma).

**Murine spermatocyte cell spread preparation**. Wildtype C57.6J/Bl6 mice were sacrificed using $CO_2$, followed by cervical dislocation. Testes were resected and seminiferous tubules extracted and immersed in PBS after decapsulation of the testes. Next, nuclear spreadings were carried out as described by de Boer et al.[52]. Briefly, seminiferous tubules were transferred to hypotonic buffer and incubated for 60 min. Individual seminiferous tubules were transferred to sterile sucrose solution on a slide, disrupted with forceps, and cells flushed out by resuspension with a 10 μl pipette. In parallel, poly-lysine slides were immersed in 1% formaldehyde, substituted with acrylamide in case of the MAP and U-ExM protocol (30% AA, 4% FA in PBS for MAP and 1% AA, 0.7% FA in PBS for U-ExM). 20 μl of the testes cells in sucrose were transferred to a drop of the formaldehyde solution collected in a corner of the slide and spread evenly across the entire slide. Slides were incubated in a wet chamber for 2 h and dried overnight in a wet chamber left ajar. To ensure ease of handling, nuclei were spread on round 18 mm coverslips (NO. 1.5H).

**Antibodies**. Guinea pig and rabbit anti-SYCP1 (N-terminal amino acids 1–124)[53], guinea pig anti-SYCP3 (N-terminal amino acids 27–38)[54], and rabbit anti-SYCE3 (full length protein)[55] were generated by Seqlab through immunizing the host with the respective peptides and affinity purified before use. Rabbit anti-SYCP3 (NB300-232; derived against the C terminus) was purchased from Novus Biologicals and mouse anti-SYCP3 (ab97672; full length protein) was purchased from Abcam. Al647 goat anti-guinea pig IgG (H + L), highly cross-absorbed (A-21450) Invitrogen (ThermoFisher); goat anti-rabbit IgG (H + L) highly cross-adsorbed Alexa Fluor 568 (A-11036) was purchased from ThermoFisher; Alexa 488 conjugated AffiPure F(ab′)2 goat anti-guinea pig IgG (H + L) was purchased from Dianova (106-546-003). SeTau647 NHS (K9-4149; SETA Biomedicals) was conjugated to F (ab′)2 of goat anti-mouse IgG (SA-10225; ThermoFisher); F(ab′)2 goat anti-rabbit IgG (H + L) cross-adsorbed Alexa Fluor 647 (A-21246) and F(ab′)2 goat anti-mouse IgG (H + L) cross-adsorbed Alexa Fluor 488 (A-11017) were purchased from ThermoFisher. Supplementary Table 1 summarizes the immunolabeling used for the different experiments.

**Immunofluorescence of non-expanded SCs**. Coverslips with fixated nuclear spreadings were washed with PBS. Unspecific epitopes were blocked for 1 h in 10% normal goat serum (NGS). Nuclear spreadings were incubated face down on 100 μl of the primary antibody for 1 h at room temperature in a humidified chamber. Spreadings were then washed with PBS and blocked for 30 min in 10% NGS before incubation with the secondary antibody for 30 min at room temperature in a humidity chamber. For multicolor experiments, immunostaining was performed sequentially, starting with the antibody raised in mouse.

**Protein-retention protocol (proExM)**. Gel linker treatment: In order to cross-link amide groups into the polymeric hydrogel network, samples were incubated in freshly prepared amine reactive AcX solution (0.1 mg/ml) in PBS. Dessicated AcX stocks (10 mg/ml) stored as aliquots at −20 °C were therefore resuspended in 10 μl anhydrous DMSO and diluted 1:100 in PBS (1×). Samples were then covered with 1 ml AcX solution per coverslip and incubated at room temperature in a humidified chamber overnight.

Gel formation, digestion, and expansion: Hydrogel formation was carried out on a cell culture plate lid covered with parafilm. The plate was placed on ice to provide a cooled and flat hydrophobic gelation surface. 80 μl of pre-chilled (4 °C) gelling solution (8.55% SA, 2.5% AA, 0.15% Bis-AA, 0.2% APS, 0.2% TEMED, 11.7% NaCl, 1× PBS) were prepared from proExM Monomer stock solution consisting of a mixture of acrylic copolymers and crosslinking agent (8.55% SA, 2.5% AA, 0.15% Bis-AA, 11.7% NaCl, 1× PBS) that was supplemented with radical polymerization initiator APS and accelertator TEMED right before use. The gel solution was placed on parafilm and coverslips were put on top of the formed droplet with spread cells facing down. After 5 min incubation on ice, crosslinking polymerization was allowed to occur for 1.5 h at 37 °C in a humidified chamber. Then samples were treated with 8 U/ml Proteinase K in Digestion Buffer (50 mM Tris pH (8.0), 1 mM EDTA, 0.5% Triton X-100, 0.8 M guanidine HCl). For expansion of the samples, hydrogels were washed several times in double-deionized water until the maximum extent of swelling of the gels was reached.

**Magnified analysis of the proteome (MAP) and ultrastructure expansion microscopy (U-ExM) protocol**. Gel linker treatment: In the case of MAP and U-ExM-expanded samples, spreads were incubated in AA/FA solution (30% AA, 4% FA in PBS for MAP and 1% AA, 0.7% FA in PBS for U-ExM) for 4 h at 37 °C in a humidified chamber before proceeding with gelation of the samples.

Gel formation, denaturation and expansion: Following AA/FA incubation, samples were washed three times for 5 min each in PBS (1×). Then polymerization of hydrogels was performed as described under 'Protein retention protocol' using an optimized MAP Gel solution (7% SA, 20% AA, 0.05% Bis-AA, 0.5% APS, 0.5% TEMED, 1× PBS) or U-ExM Gel solution (19% SA, 10% AA, 0.1% BIS-AA, 0.5% APS, 0.5% TEMED, 1× PBS). The MAP monomer solution composition was altered compared to the original recipe[7] regarding the monomer to crosslinking agent ratio. After polymerization hydrogels were carefully removed from the coverslips and transferred directly into pre-heated (95 °C) Denaturation buffer (200 mM SDS, 200 mM NaCl, 50 mM Tris, pH 9.0) in 1.5 ml centrifuge tubes. Samples were then denatured for 1 h in a heating block incubator with closed tube lids. For swelling of the sample, hydrogels were washed with excess volume of double-deionized water that was exchanged several times until the maximum expansion level was reached.

**Post-expansion immunolabeling of MAP and U-ExM-treated samples**. For MAP- and U-ExM-treated samples immunostaining was performed post-expansion. Fully expanded gels were incubated in Blocking buffer (0.15% BSA in PBS) twice for 30 min each. Gels shrink during this blocking step. Then primary antibodies were incubated sequentially for 3 h each at 37 °C in a humidified chamber with two 20 min washing steps in Blocking buffer following each antibody incubation step. Next a secondary antibodies mix was added simultaneously for 3 h at 37 °C in a humidified chamber. Samples were then washed twice for 30 min each with Washing Buffer (0.1% Tween-20 in PBS) and once more overnight. Samples were then washed in double-deionized water and expanded back to the maximum hydrogel volume.

**Mounting of expanded samples**. Poly-lysine coating of cover glasses: 24 mm round cover glasses (NO. 1.5. H) were sonicated in double-deionized water, 1 M KOH and absolute ethanol (≥99.8%) for 15 min each. Following each sonication step, glasses were rinsed with double-deionized water. After sonication glasses were finally dried at 100 °C in an oven. Cover glasses were then covered with 0.1% poly D-lysine and incubated for 1 h at room temperature. Next, glasses were washed again with water and air-dried and stored in a closed glass container to avoid dust contamination. Coated coverslips were stored at 4 °C and used for up to 1 week.

Fluorescent marker treatment of cover glasses: Fluorescent beads introduced directly into the hydrogel show strong fluorescence loss caused by persulfate radicals during polymerization. For this reason, we directly coated coverslips with fluorescent markers to perform channel alignment. Therefore, fluorescent marker stock suspension (0.1 μm, ~1.8 × 10^{11} particles/ml, TetraSpeck Microspheres, ThermoFisher) was vortexed for ~1 min and then diluted 1:1000 in PBS (1×) and vortexed again briefly. Glasses were covered with the suspension for 30 min at room temperature to let the fluorescent markers settle down. Then coverslips were washed with double-deionized water, air-dried, and thereupon used for hydrogel immobilization with Agarose.

Agarose embedding: Expanded samples were cut into ~1.5 ×1.5 cm pieces using a razor blade and excess water was removed carefully from the gels with laboratory wipes. Gels were then transferred onto poly-lysine-coated coverslips. To further avoid drifting of the sample during long-term image, acquisition gels were additionally embedded in 1% (w/v) agarose in water. Therefore, a second uncoated round 18-mm-coverslip was placed on top of the cut hydrogel and melted agarose

(~40 °C) was carefully applied around the sides of the hydrogel using a pipette. Care was taken to avoid Agarose from flowing below the hydrogel. The agarose gel was subsequently hardened at 4 °C for ~10 min and the upper coverslip was removed carefully. Double-deionized water was then added on the hydrogel to prevent dehydration during imaging.

**Imaging**. Structured illumination microscopy (SIM) imaging was performed using a Zeiss Elyra S.1 SIM imaging system consisting of an inverse Axio Observer. Z1 microscope equipped with a C-APOCHROMAT ×63 (NA 1.2) water-immersion objective and four different excitation lasers (405 nm diode, 488 nm OPSL, 561 nm OPSL, and 642 nm diode laser). Three-dimensional SIM imaging of the expanded sample was recorded on a PCO edge sCMOS (scientific complementary metal-oxide semiconductor) camera with $0.110 \mu m$ $z$-steps adjusted by an inserted Z-Piezo stage. Three rotations of the grid pattern were projected on the image plane for each acquired channel. The red fluorophore was imaged before fluorophores with lower wavelengths to minimize photobleaching. Raw data images were processed using the Zeiss ZEN software (black edition). Fluorescent beads mounted on the coverslip that were localized directly below the sample imaging area were set as lowest imaging plane and recorded with the sample for each 3D $z$-stack. For alignment and SIM processing, images containing fluorescent markers were cropped out and used to align the channels in each recorded $z$-stack via the Zeiss Zen software channel alignment tool. Rescanning confocal microscopy (RCM) imaging was conducted on a Nikon TiE inverted microscope combined with an RCM unit (Confocal.nl). RCM derives from the image-scanning principle, whereby pixel reassignment is carried out purely optomechanically[35,56]. The unit is connected to a Cobolt Skyra (Cobolt, Hübner Group) laser unit providing four excitation laser lines (405, 488, 561, and 640 nm). Images were recorded on an sCMOS camera (Zyla 4.2P, Andor) using a 60× water-immersion objective (CFI Plan APO, 1.27-NA; Nikon) and a fixed 50 μm pinhole size. The setup was operated by the microscope software NIS-Elements (version 4.6). $d$STORM acquisition of unexpanded samples were conducted on a home-built widefield imaging setup as described previously[20]. The setup consists of an inverted IX71 microscope (Olympus) equipped with an oil-immersion objective (APON 60XOTIRF, NA 1.49, Olympus). For excitation of Alexa Fluor 647, a 639 nm OPS laser diode (Genesis MX639-1000 STM; Coherent) was used in quasi-TIRF mode. To avoid drift during acquisition, a nose-piece stage (IX2-NPS, Olympus) is implemented in the microscope. The nose-piece stage mechanically connects the microscope objective with the sample holder and coverslip. Fluorescence light was collected onto an electron-multiplying charge-couple device (EM-CCD) camera (iXon Ultra 897, Andor). As photoswitching buffer 100 mM β-mercaptoethylamine in PBS (pH 7.4) supplemented with 0.1 g/ml glucose and 0.5 mg/ml glucose oxidase as oxygen scavenger system was used. For image reconstruction, the ImageJ plugin ThunderSTORM[57] was used.

**Protein position analysis**. SC cross-sectional profiles using 'Line Profiler': The SYCE3 or SYCP3 input image is convolved with a Gaussian blur, compensating noise and intensity fluctuations. Via Otsu thresholding the image is converted into a binary image[58]. Using Lee's algorithm, a skeletonize image is constructed reducing all expanded shapes to lines with 1 pixel width[59]. Subsequently, all connected pixels unequal to zero are sorted und checked for continuity, i.e. sharp edges mark a breakpoint initiating a new line. The pixel coordinates of each line are then fitted with a c-spline. The c-splines coordinates and local derivatives are a good estimation for the center and orientation of the helical structure of the SYCP3 channel. Note that the orientation and center of the SYCP3 channel can also be estimated by determining the center of mass ('1 channel' mode). Combining spline fitting of the central element with a floodfill algorithm used on the SYCP3 strand, yields potential regions of interest. Using a logical AND operation on the line coordinates and the floodfill image results in the desired source coordinates and directions for further evaluation. Line profiles are then constructed originating at the source coordinates perpendicular to the c-splines derivative. We discarded all distances laying outside of 400–1200 nm to exclude undesired structures, like two strands close to each other. As Line Profiler uses 2D images, it is biased to underestimate the true peak-to-peak distance close to helical twists. Therefore, we histogrammed the distances of all profiles and fitted them to a right-sided half norm formula (1):

$$y = \begin{cases} 0 \; for \; x - c \in [-\infty, 0] \\ Ae^{-\frac{(x-c)^2}{2\sigma^2}} + n \; else \end{cases}, \tag{1}$$

of intensity $y$ at position $x$ with amplitude $A$, center $c$, standard deviation $\sigma$, and noise level $n$ as free fit parameters. Strand distances and errors were determined by the center of the least-square fit and its standard deviation. The line profiles are post aligned at the center of the two global maxima. Average profiles are returned for each line and for the whole image.

Protein distribution of SYCE3 and SYCP1N: To determine the distribution of SYCE3, areas of the SYCE3 signal were manually chosen from regions where SYCP3 strands showed a helix crossing. These regions can be regarded as frontal views of the SYCE3 and SYCP1N proteins and were used for cross-sectional profiling (Fig. 4).

In order to demonstrate that data evaluation is reliable, we simulated a helix structure and analyzed it with Line Profile. The maximum intensity projection of a helix structure can be well described by the formula $y_{1,2} = \pm a \, \cos(bx)$ with amplitude $a$ and elongation $b$. As can be seen in unexpanded and expanded images of SYCP3, the strand distance decreases with the helix structure changing from lateral to frontal view (Supplementary Figs. 1–3). To ensure that this fact does not result in smaller peak to peak distances, we simulated a helix structure with $a = 800$ nm and $b = 1/1612$ nm$^{-1}$ and rendered it with a pixel size of 32.24 nm. The helical structure was then evaluated with Line Profiler. The determined strand distance of $800 \pm 5.0$ nm verifies that our evaluation is not biased by the decreasing strand distance (Supplementary Fig. 13).

Averaging of cross-sectional profiles: The resulting cross-sectional profiles were averaged using the data analysis software Origin(Pro) Version 2016.

Expansion factor determination and analysis of expansion isotropy: To compare pre- and post-expansion images, spreaded spermatocytes on 18 mm coverslips were labeled pre-expansion with primary antibodies against SYCP3 and secondary Alexa Fluor 488 conjugated antibodies. The central region of the coverslip was scanned in a square of ~10 × 10 mm$^2$ using a ×10 air-objective. The overlapping RCM image tiles were aligned to a large overview image of the sample and served as a map to find the same cells post-expansion. Pre-expansion RCM image z-stacks were recorded from several cells using a ×60 water-immersion objective. To determine the expansion factor, the same cells were recorded post-expansion after treating the cells according to the MAP protocol with post-expansion labeling of SYCP3 with SeTau647 conjugated secondary IgG antibodies. Corresponding pre- and post-expansion images were then registered using a rigid registration using the open source command-line program elastix as described in detail in literature[5]. To investigate deformations caused by the expansion process, a deformation vector field of pre- and post-expansions images were created using elastix and transformix[60,61].

3D visualization of MAP-SIM data: 3D-MAP-SIM data shown in Supplementary Movie 1 and Supplementary Movie 2 were visualized using the microscopy image analysis software IMARIS (version 8.4.1, Bitplane).

**Reporting summary**. Further information on research design is available in the Nature Research Reporting Summary linked to this article.

## Data availability

All the data supporting the findings of this study are available from the corresponding authors upon reasonable request.

## Code availability

The automated image-processing software Line Profiler is available at https://line-profiler.readthedocs.io/en/latest/.

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

## Acknowledgements

We thank Dominic Helmerich for technical assistance with the SeTau647 labeled antibody. We are very grateful to K. Schücker for the provision of raw data. This work was supported by the German Research Foundation (SA829/19-1, TRR 166 Receptor Light project A04, and Be1168/8-1). M.-C.S. was supported by a Boehringer Ingelheim Fonds Travel Grant and a HHMI Grant to attend the 2018 CSHL course 'Quantitative Imaging: From Acquisition to Analysis' and wishes to thank the faculty of the course for excellent training and illuminating discussion benefitting the preparation of this manuscript-specifically Jennifer Waters, Talley Lambert, Hunter Elliot, and Suliana Manley, as well as Marcelo Cicconet, Jessica Hornick, Anna Jost, and Michael Weber.

## Author contributions

F.U.Z., M.C.S., M.S., and R.B. conceived and designed the project. M.S. and R.B. supervised the project. F.U.Z., M.C.S., A.K., and T.K. performed all experiments. S.R. developed Line profiler. S.R. and F.U.Z. performed data analysis. All authors wrote and revised the final manuscript.

## Competing interests

The authors declare no competing interests.
