## [Peer Review File · Nature Communications]

Reviewers' comments:

Reviewer #1 (Remarks to the Author):

This manuscript presents a comparison between different high-resolution microscopy modalities involving sample expansion, using the well-characterized dimensions of the synaptonemal complex as a standard. Through innovations in many steps of sample and gel preparation, the authors demonstrate that expansion microscopy followed by 3D structured illumination results in optimal sample preservation, resolution equivalent to the best non-expanded single-molecule localization-based microscopy, and the ability to image 3 (or more) color channels. The synaptonemal complex images are truly stunning, and compare with some of the best light and electron microscopy images yet taken.

Of special note are the clear recapitulation of axis fraying (resolved well by MAP-SIM and shown to be SYCP3-positive) and occasional doubling observed of single lateral elements. The ability to observe these rare and very small-scale structural features with immunofluorescence is very exciting, and demonstrates that methods such as MAP-SIM are poised to open some new areas of investigation in cell biology.

While the images speak for themselves to the high quality of the methods, the structure of the manuscript itself is often not as clear. Figures 1–3 are straightforward and mostly solid recapitulations of previous work, with the exception of the novel light-microscopy visualization of split signals of a single SYCP3 axis in 2f. _It would be very informative if we were told how often and under what circumstances this phenomenon can be visualized._

For the other figures, it is unfortunately often difficult to assess the connection between the figure and the text. The main recurring problem is that figure call-outs in the main text tend to lack descriptions and subpanel labels (i.e., not just "Figure 4" but "Figure 4b" is better when appropriate) to aid the reader. Figure 4, for example, is only referred to once, in a rather general way, in the main text, although it has 7 subpanels that make different points between them. Readers are left to figure it out for themselves, which makes the paper somewhat challenging to read in parts.

Major comments:

The whole section from lines 260-286 is rather difficult to understand as it is written; further, it makes the claim "MAP-SIM discloses new information about the molecular organization of the SC" but does not say clearly what this new information is. Stating that your method "indicated a far more complex distribution of the central region proteins" does not tell me what is new. This section should be rewritten with a focus on concretely describing what has been found out, to the extent this is possible.

Within this section, in lines 273–274: "the signal distributions of the MAP-SIM imaged SYCP1N-terminus and SYCE3 indicated a far more complex distribution of the central region proteins than the bimodal distribution previously described by dSTORM and immunogold EM" -- although no figure panel is mentioned, I assume this is referring to the image in Figure 4c (unfortunately obscured by the unnecessary red SYCP3 signals) and the corresponding bimodal pattern of profile #2 in Figure 4d. However, given the small sample size here (one profile), the novelty of the method, and the fact that the averaged profiles show a clearly monomodal distribution of SYCP1N and SYCE3, is it possible that this bimodal profile is an artifact/noise? It may not be, but some evidence should be provided or at least the possibilities discussed.

In Fig. 4e's legend: "Frontal views of the two proteins used for cross-sectional profiles are shown in yellow" - it is not clear what this means; does this mean that yellow is a false-color overlay on the regions selected for LineProfiler analysis? I suggest using arrows or brackets to show these regions, instead of a false color, which prevents a clear view of the image itself (for instance, using magenta/green alone would allow an appreciation of the SYCE1/SYCP1N signals in the absence of

the red SYCP3 signals from a,b, and c). Even better would be to point out these regions in 4a so we can see where they lie with respect to the twisting of the lateral elements.

Since the LineProfiler analysis seem to be 2d only, some discussion should be included in the text of how this could tend to underestimate the true peak-to-peak separation, since helical twisting of lateral elements will bring their XY projections closer together than their true distance. Some part of this discussion is found in the legends for supplemental Figure 2, but perhaps it is an important enough point to be mentioned in the main text.

Minor comments:

For the LineProfiler analysis, please clarify at what point the input image is reduced from 3D to 2D - is this working on sections or projections, and if so what kind of projections, etc.

Line 137: I don't think it's clear that lateral elements directly bind to chromatin, nor that central elements bind to lateral elements. Figure 1's legend states that the transverse filaments "connect" lateral and central elements, but this is not known and may or may not be a universal feature.

Line 154: "...revealed that the lateral element protein SYCP3 shows a bimodal distribution separated by 221.6 ± 6.1 nm (SD) (Fig. 2)" -- but Fig. 2 does not show this value, nor a bimodal distribution.

Line 197 refers to U-ExM-induced SC structural breaks shown in Sup. Figure 4, but which panel within figure 4 shows the breaks is not mentioned, nor are the breaks highlighted or mentioned within the figure; please annotate this so the reader can follow the text. Likewise for lines 200-201, "pre-expansion labeling resulted in a lower labeling density due to irreversible fluorophore destruction during free-radical polymerization (Supplementary Figs. 4 and 5)", but these figures do not point this out, in the figure itself or the legends.

From line 211, "the main difference between centrioles and SCs is their biomolecular composition" until line 219, the text unfortunately becomes quite difficult to understand. I think this part should be rewritten and placed in the discussion, since it appears to be rationalizing why these results were different from previous results with centrioles.

Line 230 -- re: previous dSTORM experiments showing C termini at the lateral elements and N termini in the center — it would be good to point out that immunoEM also showed this in the mid-90's (Liu, J. G. et al. Localization of the N-terminus of SCP1 to the central element of the synaptonemal complex and evidence for direct interactions between the N-termini of SCP1 molecules organized head-to-head. *Exp. Cell Res.* 226, 11–19 (1996).)

Lines 261-270: if you are using terms like "lateral" and "frontal" to describe the views of the SC, please provide an explanatory diagram in an early figure (figure 1 would be a good place) where these are defined, otherwise the reader may not understand what is meant.

Lines 275–278: I am not aware of evidence that shows the central element is particularly susceptible to competition effects in immunolabeling due to its density; if a reference could be provided, please do so; otherwise, if this is speculative, it should not be presented as fact.

Fig. 5e, please indicate the XY pair with arrows (it may not be obvious to everyone).

Several mentions are made of "twisted areas" or "places where the lateral elements twist"; but since the lateral elements are twisting everywhere to some extent, it would be good to make sure the reader understands that it's not the case that axes only twist at places that can be observed as crossings in the XY plane.

Line 447: "as described under 'proexm protocol'", but there is no such subheading—should be written "Protein retention protocol".

Line 520: "To avoid drift during acquisition a nose-piece stage (IX2-NPS, Olympus) is implemented in the microscope." -- this needs more detail; a catalog number doesn't describe in a reproducible way how drift is avoided.

Line 524: "weight per volume" could be more precisely expressed as grams / milliliters (if this is the case).

Peter Carlton, Kyoto University

Reviewer #2 (Remarks to the Author):

Review:

In Zwettler et al., the authors developed an optimized expansion microscopy protocol for visualizing the synaptonemal complex. To do so, the authors tested and optimized several expansion microscopy variants for both isotropy, labeling density with respect to SC ExM. They demonstrated 3 color SIM together with an optimized version of MAP, an ExM variant. The manuscript is well written, detailed and data from experiments is clear. I think the authors could make a stronger point about the significance of the biological discovery, as well as the methodological improvements, and how does it apply to those outside of SC biology.

I think the authors could address the following main points before recommendation for publication at Nature Comms:

Major comments:

Can the authors clarify why the expansion factors used for different versions of the ExM protocol. The expansion factor in ExM can be tuned with the crosslinker density, for proper comparison of labeling density: all protocols be compared at ~ same expansion factor of 3 between ExM, MAP and U-ExM?

The authors state that "post-expansion labeling protocols substantially increases the epitope accessibility for IgG antibodies resulting in higher labeling densities especially of sterically demanding multiprotein complexes"

This is a significant claim and perhaps one of the most important takeaways from the paper, as super-resolution imaging is fundamentally limited by labeling density. Can the authors provide quantitation to this point?

Specifically, does ExM-SIM offer higher labeling densities than STORM in 1 color (STORM is prelabeled). If so, can higher resolutions be achieved, or are there more continuous structures. Can a labeling density metric be derived?

The authors make several comparisons with EM, stating that MAP-SIM resolves similar structures (i.e. SYCP3 fraying). Since MAP-SIM has molecular resolution, as well as multiple colors, have the authors been able to discover aspects of the molecular architecture that was previously unknown?

Minor comment:

The authors developed a line cut tool and made it broadly available, which is very commendable. It's unclear if computational tool is generally useful for other applications beyond the SC complex.

Reviewer #3 (Remarks to the Author):

Zwettler et al reports various expansion microscopy (ExM) methods combined with structured illumination microscopy (SIM) to investigate the molecular architecture of the synaptonemal complex (SC). The authors developed an image processing software for unbiased generation of line profiles from super-resolution images of the synaptonemal complexes. By comparing the established structural data from dSTORM, three different expansion methods of ProExM, U-ExM and MAP were evaluated in terms of expansion factors and labeling densities. MAP-SIM with the best labeling density was used for revealing new structural features of SCs.

Since ExM allows for super-resolution imaging with robust commercial instruments, many

expansion protocols has been introduced rapidly. However, for ExM to address new, unknown structures, it is crucial to precisely quantify the value and uniformity of the expansion factor. This paper provides a reference structure and automated software that are suited for providing such evaluation platform for ExM. However, the current version of manuscript do not provide sufficient supporting data for validating the performance of the software, the isotropic expansion of the MAP protocol and their new findings on SC structures from 3-color SIM imaging. For publication, I request for more supporting data that I explain in detail as follows.

1. What are the criteria that "Line Profiler" software choose the area for generating line profiles for measuring SC widths and for generating averaged profiles in Supp Figure 1-3 and 5? In other words, how did the authors select the white crossing lines in Figs. 4f, Supplementary Fig. 1a-b, 2a-b and 3a/c? The software output image in Supp Fig. 6 suggests that the software make line profiles everywhere along the SC. But, somehow the white lines that I mentioned indicate that subsets of line profiles were used to measure peak-to-peak distances and to generate average profiles. The selection procedure could impose bias to the statistical results, thus the details should be described in the paper.

2. Why do the average values of SYCP3 strand distances from 2-channel and 1-channel analyses in Supp Figure 1 and 2 differ so much (580/607 nm vs. 632/635 nm)? Since the sample preparation is the same, the expansion factor should be the same. Why did the authors used the value from 1-channel analysis results to calculate the expansion factor? If 1-channel analysis is more precise or more reproducible, evaluate the performances of the two different analyses and explain why their results differ.

3. The authors claim "isotropic" expansion by simply showing the images. I cannot judge the isotropicity by visual inspection of Fig. 3. ExM papers often show transformation matrix by overlapping pre- and post-expansion images. I cannot find such quantitative measures in this paper. Or, instead, since the authors developed the image processing software, the distributions of SYCP3 strand distances can be used to quantitatively evaluate the isotropicies of proExM, U-ExM and MAP at the ultrastructural level. Since the expansion factors differ for proExM, U-ExM and MAP, I suggest to normalize each distribution by the corresponding expansion factor for comparison of the three expansion protocols.

4. From a single profile example in Fig. 4d, the authors claimed "far more complex distribution of the central region proteins than the bimodal distribution". Simply saying "more complex" is not helpful to figure out the differences. Describe more specifically how the distribution differ from bimodal. Also, prove the higher complexity with more data. One example image is not enough.

5. The only obviously new structures in the paper is the bifurcation and fraying of the two SC strands in Fig. 5. I am not an expert in meiosis to judge the importance of the findings. The authors discussion on these structures sounds descriptive to me. What would be the biological, functional implications of these new findings?

Followings are minor comments.

a) The following sentence in the introduction does not make sense.

"The spreading of the SCs directly onto the surface of the coverslip results in the localization of the proteins close to the coverslip post-expansion and allows the use of a water-immersion objective for imaging of hydrogels without spherical aberrations."

If the sample is placed close to the surface, even oil immersion lens can be used. Did the authors mean that high NA lenses can be used?

b) In the last paragraph of the introduction, the authors claim the resolution of ExSTORM is 20-30 nm, which is typical resolution for unexpanded STORM. The ExSTORM paper claims for 10-20 nm resolution. Do the authors of this paper have evidences supporting that ExSTORM resolution is worse than reported?

c) High-resolution STORM may be limited in two colors. But, three-color DNA-PAINT is possible with exchangePAINT and offers even higher resolution than STORM. What would be the

advantages of 3-color MAP-SIM over 3-color DNA-PAINT?

d) Online Methods, "Immunofluorescence of non-expanded SCs": What is NGS? Normal goat serum?

e) Figure 3. Caption: There is no "g" panel in the figure. Perhaps, the "g" in the caption is typo for "f".

Point-by-point response to the referees' comments:

We wish to thank the reviewers for their positive and constructive comments that have helped to improve the manuscript and strengthen its novelty, performances and the fields of applications. We have revised the manuscript in depth taking into account all the remarks.

Reviewer #1 (Remarks to the Author):

This manuscript presents a comparison between different high-resolution microscopy modalities involving sample expansion, using the well-characterized dimensions of the synaptonemal complex as a standard. Through innovations in many steps of sample and gel preparation, the authors demonstrate that expansion microscopy followed by 3D structured illumination results in optimal sample preservation, resolution equivalent to the best non-expanded single-molecule localization-based microscopy, and the ability to image 3 (or more) color channels. The synaptonemal complex images are truly stunning, and compare with some of the best light and electron microscopy images yet taken.

Of special note are the clear recapitulation of axis fraying (resolved well by MAP-SIM and shown to be SYCP3-positive) and occasional doubling observed of single lateral elements. The ability to observe these rare and very small-scale structural features with immunofluorescence is very exciting, and demonstrates that methods such as MAP-SIM are poised to open some new areas of investigation in cell biology.

We thank the reviewer for his positive remarks and suggestions to further improve the quality of our manuscript.

While the images speak for themselves to the high quality of the methods, the structure of the manuscript itself is often not as clear. Figures 1–3 are straightforward and mostly solid recapitulations of previous work, with the exception of the novel light-microscopy visualization of split signals of a single SYCP3 axis in 2f. It would be very informative if we were told how often and under what circumstances this phenomenon can be visualized.

We thank the reviewer for careful reading of our manuscript. The phenomenon signal splitting of the SYCP3 axis has been observed frequently in MAP-SIM images of SC complexes. We prepared two new Supporting Figures, Supplementary Figure 11 and Supplementary Figure 12, that highlight regions with signal splitting at several SC's. The splitting could most often be observed in regions before crossing overs of the SYCP3 filaments where the structure rotates towards the lateral view. Here the signal is becoming thicker and a partial splitting of the signal can be observed. To better highlight the splitting, we included single z-slices of the image stacks in the figures and also show the course of two strands turning from frontal towards lateral view in detail in Supplementary Figs. 11 and 12 in the revised version of our manuscript.

For the other figures, it is unfortunately often difficult to assess the connection between the figure and the text. The main recurring problem is that figure call-outs in the main text tend to lack descriptions and subpanel labels (i.e., not just "Figure 4" but "Figure 4b" is better when appropriate) to aid the reader. Figure 4, for example, is only referred to once, in a rather general way, in the main text, although it has 7 subpanels that make different points between them. Readers are left to figure it out for themselves, which makes the paper somewhat challenging to read in parts.

We fully agree with the reviewer and added the required information in the revised version of our manuscript to improve readability especially with regard to Figure 4.

Major comments:

The whole section from lines 260-286 is rather difficult to understand as it is written; further, it makes the claim "MAP-SIM discloses new information about the molecular organization of the SC" but does not say clearly what this new information is. Stating that your method "indicated a far more complex distribution of the central region proteins" does not tell me what is new. This section should be rewritten with a focus on concretely describing what has been found out, to the extent this is possible.

Within this section, in lines 273–274: “the signal distributions of the MAP-SIM imaged SYCP1N-terminus and SYCE3 indicated a far more complex distribution of the central region proteins than the bimodal distribution previously described by dSTORM and immunogold EM” -- although no figure panel is mentioned, I assume this is referring to the image in Figure 4c (unfortunately obscured by the unnecessary red SYCP3 signals) and the corresponding bimodal pattern of profile #2 in Figure 4d. However, given the small sample size here (one profile), the novelty of the method, and the fact that the averaged profiles show a clearly monomodal distribution of SYCP1N and SYCE3, is it possible that this bimodal profile is an artifact/noise? It may not be, but some evidence should be provided or at least the possibilities discussed.

We thank the reviewer and re-wrote the section to describe our findings in more detail. Indeed the description was confusing and misleading. The bimodal distribution of the central element proteins SYCE3 and the N-terminus of SYCP1 has been seen in several lateral view MAP-SIM images. We therefore added Supplementary Fig. 10 in the revised version of our manuscript to support the generality of findings.

We revised the text accordingly. It reads now:

“Therefore, we next analyzed the signal distribution of the N-terminus of the transverse filament protein SYCP1 by MAP-SIM of nuclear spreadings in more detail. In addition to analyzing the distribution of proteins in frontal views, we also generated line profiles of SC proteins from lateral views (Figs. 4a-c, Supplementary Fig. 10, and Supplementary Movie 1). In agreement with EM tomographic data³⁸, the signal distributions of the MAP-SIM imaged SYCP1 N-terminus and SYCE3 indicated a far more complex distribution of the central region proteins than the bimodal distribution previously described by dSTORM and immunogold EM^{19,20}. In some lateral view sections of MAP-SIM images of the SYCP3 signal the N-terminus of the transverse filament protein SYCP1 and the central element protein SYCE3 revealed bi- and multimodal signal distributions supporting a multilayered organization of the central element (Fig. 4d and Supplementary Fig. 10). On the other hand, averaged cross-sectional intensity profiles of SYCE3 and SYCP1N (Figs. 4e,f) clearly showed a monomodal distribution of the two proteins (Fig. 4g and Supplementary Fig. 10) in accordance with the unstructured organization of the transverse filaments into a large single layer as revealed by EM tomography in a previous study²⁷. The occasional disclosure of multimodal distributions in some lateral view sections of MAP-SIM expanded SCs can be attributed to the high labeling density provided by MAP. Combined with the post-processing super-resolution of SIM, MAP-SIM enables even the resolution of individual SYCP1 positive transverse filaments which is reflected in the few detected multimodal distributions..

In Fig. 4e’s legend: “Frontal views of the two proteins used for cross-sectional profiles are shown in yellow” - it is not clear what this means; does this mean that yellow is a false-color overlay on the regions selected for LineProfiler analysis? I suggest using arrows or brackets to show these regions, instead of a false color, which prevents a clear view of the image itself (for instance, using magenta/green alone would allow an appreciation of the SYCE1/SYCP1N signals in the absence of the red SYCP3 signals from a,b, and c). Even better would be to point out these regions in 4a so we can see where they lie with respect to the twisting of the lateral elements.

We thank the reviewer for this constructive reference and revised Figure 4 accordingly. In particular, we removed the false color yellow in regions used for cross-sectional profiles and added white dotted boxes to mark these regions. Additionally we added an inset showing the crossing points of the lateral element with arrows pointing at the crossing points.

Since the Lineprofiler analysis seem to be 2d only, some discussion should be included in the text of how this could tend to underestimate the true peak-to-peak separation, since helical twisting of lateral elements will bring their XY projections closer together than their true distance. Some part of this discussion is found in the legends for supplemental Figure 2, but perhaps it is an important enough point to be mentioned in the main text.

We agree with the reviewer that this is an important issue. Since Lineprofiler uses 2D-images it is biased to underestimate the true peak-to-peak distance close to helical twists. Therefore, we evaluated the strand distance by fitting a right sided half norm formula to the histogrammed distances taken by Lineprofiler. We added a simulation of a helical structure with a predefined strand distance (see Methods). Evaluating this structure with Lineprofiler and the subsequent histogram fit, yields exactly the given strand distance (new Supplementary Fig. 13).

Minor comments:

For the LineProfiler analysis, please clarify at what point the input image is reduced from 3D to 2D - is this working on sections or projections, and if so what kind of projections, etc.

LineProfiler is using maximum intensity projections. We added the missing information to the Methods part in the revised version of the manuscript.

Line 137: I don't think it's clear that lateral elements directly bind to chromatin, nor that central elements bind to lateral elements. Figure 1's legend states that the transverse filaments "connect" lateral and central elements, but this is not known and may or may not be a universal feature.

We agree with the reviewer that our phrasing here is too general. It now reads "In mouse, the central region is formed by a central element running between the lateral elements, and numerous transverse filaments connecting the lateral elements and the central element (Fig. 1)." In accordance, we changed the caption of figure 1 to "The murine synaptonemal complex".

Line 154: "...revealed that the lateral element protein SYCP3 shows a bimodal distribution separated by 221.6 ± 6.1 nm (SD) (Fig. 2)" -- but Fig. 2 does not show this value, nor a bimodal distribution.

We thank the reviewer for careful reading. We agree the given value of 221.6 ± 6.1 nm (SD) for the distance between the lateral element protein SYCP3 comes from our previous dSTORM publication (Ref. 20). The reference to Figure 2 is confusing and was therefore removed.

Line 197 refers to U-ExM-induced SC structural breaks shown in Sup. Figure 4, but which panel within figure 4 shows the breaks is not mentioned, nor are the breaks highlighted or mentioned within the figure; please annotate this so the reader can follow the text. Likewise for lines 200-201, "pre-expansion labeling resulted in a lower labeling density due to irreversible fluorophore destruction during free-radical polymerization (Supplementary Figs. 4 and 5)", but these figures do not point this out, in the figure itself or the legends.

We thank the reviewer for his constructive reference and revised Supplementary Figure 4 accordingly. That is, we added an expanded view highlighting the structural breaks induced in U-ExM

samples and specified the panels in the text. A lower label density was found with the proExM protocol due to the need of very high laser powers to excite fluorophores during SIM measurements (in contrast to post-labeling methods that worked well with laser powers that are also used for unexpanded samples). As different fluorophores and laser powers were used for the different expansion protocols an objective comparison of intensities is not feasible. For these reasons we removed the text parts stating a lower labeling density.

From line 211, "the main difference between centrioles and SCs is their biomolecular composition" until line 219, the text unfortunately becomes quite difficult to understand. I think this part should be rewritten and placed in the discussion, since it appears to be rationalizing why these results were different from previous results with centrioles.

We agree and revised this part of the manuscript to improve the comprehensibility.

Line 230 -- re: previous dSTORM experiments showing C termini at the lateral elements and N termini in the center — it would be good to point out that immunoEM also showed this in the mid-90's (Liu, J. G. et al. Localization of the N-terminus of SCP1 to the central element of the synaptonemal complex and evidence for direct interactions between the N-termini of SCP1 molecules organized head-to-head. *Exp. Cell Res.* 226, 11–19 (1996).)

We thank the reviewer for his remark and added the reference in the revised version of the manuscript.

Lines 261-270: if you are using terms like "lateral" and "frontal" to describe the views of the SC, please provide an explanatory diagram in an early figure (figure 1 would be a good place) where these are defined, otherwise the reader may not understand what is meant.

We agree with the reviewer that the reader will benefit from a visual reference to the lateral and frontal view sections of the SC. Therefore, we added an explanatory diagram to Figure 1 in our revised manuscript that shows the frontal and lateral view sections of the SC (Fig. 1b).

Lines 275–278: I am not aware of evidence that shows the central element is particularly susceptible to competition effects in immunolabeling due to its density; if a reference could be provided, please do so; otherwise, if this is speculative, it should not be presented as fact.

We thank the reviewer for his remark. Several studies mention potential limited epitope accessibility of central element proteins (Hernandez-Hernandez et al., 2016 and Schücker et al., 2015). However, since this limited epitope accessibility remains speculative, we have rephrased lines 275-278 to "The central element is on average 26 nm by 38 nm and comprises at least seven proteins in the mouse. In traditional immunofluorescence and immuno-EM preparations antibody-antibody-fluorophore and antibody-antibody-gold complexes localize in a radius of 20-25 nm around the central antigen³⁶⁻³⁹. After binding of the first complexes to their respective epitopes, free epitopes at the core of the central element are likely difficult to access. Using MAP-SIM epitope accessibility is improved due to ..".

Fig. 5e, please indicate the XY pair with arrows (it may not be obvious to everyone).

We have added arrowheads to figure 5e indicating the XY pair.

Several mentions are made of "twisted areas" or "places where the lateral elements twist"; but since the lateral elements are twisting everywhere to some extent, it would be good to make sure the

reader understands that it's not the case that axes only twist at places that can be observed as crossings in the XY plane.

We fully agree with the reviewer that the SC overall is a twisted structure and referring to lateral view sections of the SC as twisted is therefore confusing. Consequently, we replaced "twisted areas" by lateral view sections. We additionally refer to the newly added explanatory diagram in figure 1b showing the lateral view of the SC.

Line 447: "as described under 'proexm protocol'", but there is no such subheading—should be written "Protein retention protocol".

We thank the reviewer and changed the sentence accordingly.

Line 520: "To avoid drift during acquisition a nose-piece stage (IX2-NPS, Olympus) is implemented in the microscope." -- this needs more detail; a catalog number doesn't describe in a reproducible way how drift is avoided.

We added the requested information. "The nose-piece stage mechanically connects the microscope objective with the sample holder and coverslip."

Line 524: "weight per volume" could be more precisely expressed as grams / milliliters (if this is the case).

We thank the reviewer for his remark and changed the concentration to "0.1 g/ml".

Peter Carlton, Kyoto University

Reviewer #2 (Remarks to the Author):

Review:

In Zwettler et al., the authors developed an optimized expansion microscopy protocol for visualizing the synaptonemal complex. To do so, the authors tested and optimized several expansion microscopy variants for both isotropy, labeling density with respect to SC ExM. They demonstrated 3 color SIM together with an optimized version of MAP, an ExM variant. The manuscript is well written, detailed and data from experiments is clear. I think the authors could make a stronger point about the significance of the biological discovery, as well as the methodological improvements, and how does it apply to those outside of SC biology. I think the authors could address the following main points before recommendation for publication at Nature Comms:

We thank the reviewer for his/her positive evaluation of our manuscript and remarks, which we fully considered in our revision to further improve the quality of manuscript.

Major comments:

Can the authors clarify why the expansion factors used for different versions of the ExM protocol. The expansion factor in ExM can be tuned with the crosslinker density, for proper comparison of labeling density: all protocols be compared at ~ same expansion factor of 3 between ExM, MAP and U-ExM?

We thank the reviewer for his/her remark and agree that it is necessary to adjust the protocols to the same expansion level for a better comparability. In fact we optimized the MAP protocol to an expansion factor of ~4x by reducing the cross-linker density. This adjustment was performed by measuring the size of the gel using calipers (which is not very precise). To confirm this expansion factor we performed a new experiment comparing correlated pre- and post-expansion images of spread spermatocytes. (see new Supplementary Fig. 6) We then calculated the physical magnification by rigid registration of several pre- and post-expansion images and analyzed these images for distortions introduced by the expansion process. (see new Supplementary Fig. 7).

Our new experiments confirmed a 4.2x expansion factor of the hydrogel with low distortions. The 4x expansion factor seems at first misleading if you compare the physical expansion with the molecular expansion factor determined by our LineProfiler analysis. This mismatch can be explained by the smaller linkage error and improved epitope accessibility which can be achieved using post-labeling expansion protocols like MAP-SIM. We also added a discussion and new Supplementary Fig. 8 to discuss and schematically illustrates the underlying difference between pre- and post-labeling which result in different expansion factors for post-expansion labeling protocols such as MAP.

The authors state that “post-expansion labeling protocols substantially increases the epitope accessibility for IgG antibodies resulting in higher labeling densities especially of sterically demanding multiprotein complexes”

This is a significant claim and perhaps one of the most important takeaways from the paper, as super-resolution imaging is fundamentally limited by labeling density. Can the authors provide quantitation to this point?

Specifically, does ExM-SIM offer higher labeling densities than STORM in 1 color (STORM is pre-labeled). If so, can higher resolutions be achieved, or are there more continuous structures. Can a labeling density metric be derived?

We fully agree with the reviewer that our finding of the advantage of post-expansion labeling regarding labeling efficiency especially of sterically demanding multiprotein complexes is one of the most important results of our manuscript. The comparison of labeling density of unexpanded versus expanded samples is, however, a very complex endeavor. First, the antibody binding constant can differ after protein denaturation as used by MAP. Hence, the binding efficiency of the same antibody can vary before and after expansion. Second, SIM is per se not single-molecule sensitive as (d)STORM is thus rendering the direct comparison difficult. In fact, only the labeling continuity and signal density along a filamentous structure like the SC can be compared, and this is what we did to compare the different expansion protocols. Due to this reasons the advantage of post-expansion labeling might be less noticeable in most experiments. Only in sterically demanding very protein dense regions of multiprotein complexes the epitope accessibility might be improved upon expansion, presupposed that the epitopes survive denaturation and expansion and are still recognized by the antibody. In the case of the SC the multilayered organization of proteins in the central element and SYCP3 fraying of the axes represent such cases where post-expansion labeling outperforms in some regions pre-expansion labeling. To cut a long story short, we fully agree with the reviewer and acknowledge his/her ideas to further increase the impact of our manuscript but the question is presently too complex to be answered correctly.

The other question whether post-expansion labeling can achieve a higher resolution has to be answered with yes mainly due to the fact that the linkage error (the distance between the fluorophore and the protein of interest) decreases substantially when using post-expansion labeling. If as in our studies, proteins are labeled by immunolabeling pre-expansion the linkage error is ~ 20 - 30 nm (primary and secondary IgG antibody with a size of 10 - 15 nm). This linkage error will “expand” during expansion – in our case ~ 4 fold to 80 - 120 nm whereas post-expansion labeling will reduce the ‘true’ linkage error to ~ 5 - 8 nm for a 4 -fold expanded sample. We highlighted these findings in more detail in the revised version of our manuscript.

The authors make several comparisons with EM, stating that MAP-SIM resolves similar structures (i.e. SYCP3 fraying). Since MAP-SIM has molecular resolution, as well as multiple colors, have the authors been able to discover aspects of the molecular architecture that was previously unknown?

The reviewer is raising a few interesting but also very challenging questions. Our data demonstrate that MAP-SIM can be used advantageously for isotropic expansion of SCs under full preservation of its molecular architecture. Furthermore, our data show that post-expansion labeling enables the visualization of structural details of protein dense regions of the SC, e.g. the multilayered organization of proteins in the central element and SYCP3 fraying of the axes, which has not been seen by light microscopy before. Of course, there are many more details of the molecular organization of SCs to be discovered possibly by a combination of expansion and super-resolution microscopy. For example, $10\times$ expansion combined with post-expansion labeling should enable real molecular resolution imaging in the near future and thus allow us to fully resolve the molecular architecture of SCs. Our manuscript demonstrates that careful optimization of expansion and post-expansion labeling protocols form the basis for reliable structural investigations of multiprotein complexes.

Minor comment:

The authors developed a line cut tool and made it broadly available, which is very commendable. It’s unclear of computational tool is generally useful for other applications beyond the SC complex.

We thank the reviewer for this comment. Line Profiler can be used for the analysis of all filamentous structures. For example, it has been used successfully to determine the spatial resolution achieved by dSTORM imaging of microtubules. Here, the cross-sectional profiles of selected microtubule areas are often consulted. If the two-dimensional (2D) projection of the fluorescence intensity distribution measured from microtubule filaments shows two separate lines, the peak-to-peak distance can then

be fitted with a sum of two Gaussians and used as an estimate of the spatial resolution. To ensure an objective evaluation and comparison of the spatial resolution achieved, we used 'Line Profiler'. A general application on expanded structures would be possible if the evaluation platform is extended with a reduction algorithm, like canny edge.

Reviewer #3 (Remarks to the Author):

Zwettler et al reports various expansion microscopy (ExM) methods combined with structured illumination microscopy (SIM) to investigate the molecular architecture of the synaptonemal complex (SC). The authors developed an image processing software for unbiased generation of line profiles from super-resolution images of the synaptonemal complexes. By comparing the established structural data from dSTORM, three different expansion methods of ProExM, U-ExM and MAP were evaluated in terms of expansion factors and labeling densities. MAP-SIM with the best labeling density was used for revealing new structural features of SCs.

Since ExM allows for super-resolution imaging with robust commercial instruments, many expansion protocols has been introduced rapidly. However, for ExM to address new, unknown structures, it is crucial to precisely quantify the value and uniformity of the expansion factor. This paper provides a reference structure and automated software that are suited for providing such evaluation platform for ExM. However, the current version of manuscript do not provide sufficient supporting data for validating the performance of the software, the isotropic expansion of the MAP protocol and their new findings on SC structures from 3-color SIM imaging. For publication, I request for more supporting data that I explain in detail as follows.

We thank the reviewer for careful reading of our manuscript and his/her valuable comments, which we fully considered in our revised version.

1. What are the criteria that "Line Profiler" software choose the area for generating line profiles for measuring SC widths and for generating averaged profiles in Supp Figure 1-3 and 5? In other words, how did the authors select the white crossing lines in Figs. 4f, Supplementary Fig. 1a-b, 2a-b and 3a/c? The software output image in Supp Fig. 6 suggests that the software make line profiles everywhere along the SC. But, somehow the white lines that I mentioned indicate that subsets of line profiles were used to measure peak-to-peak distances and to generate average profiles. The selection procedure could impose bias to the statistical results, thus the details should be described in the paper.

We agree with the reviewer and revised the text (Methods) accordingly to describe in more detail the procedure. We agree that the previous evaluation might introduce a bias to smaller peak to peak distances. Therefore, we evaluated the strand distance by fitting a right sided half norm to the histogrammed distances taken by Line Profiler. We added a simulation of a helical structure with a predefined strand distance (see Methods of revised manuscript). Evaluating this structure with Line Profiler and the subsequent histogram fit, yields exactly the given strand distance (see new Supplementary Fig. 13).

2. Why do the average values of SYCP3 strand distances from 2-channel and 1-channel analyses in Supp Figure 1 and 2 differ so much (580/607 nm vs. 632/635 nm)? Since the sample preparation is the same, the expansion factor should be the same. Why did the authors used the value from 1-channel analysis results to calculate the expansion factor? If 1-channel analysis is more precise or more reproducible, evaluate the performances of the two different analyses and explain why their results differ.

The reviewer is right, the two channel mode using the SYCP3 and SYCE3 signals outperforms one channel analysis. It is in general more accurate since the one-channel method determines the gradient solely from the signals of two SYCP3 strands, which rotate due to SCs helical structure. Utilizing in addition the signal of a central element protein (e.g. SYCE3 or SYCP1N) delivers generally a better approximation for the overall direction of the strand. However, since the reference *d*STORM experiments for calculation of the experimental expansion factor have been performed with SCs labeled solely with for SYCP3 we used the one-channel mode in our manuscript. Using the new data analysis method described in the revised version of the manuscript (Methods), 1- and 2-channel analysis provided similar results (see Supplementary Figs. 1 and 2).

3. The authors claim “isotropic” expansion by simply showing the images. I cannot judge the isotropicity by visual inspection of Fig. 3. ExM papers often show transformation matrix by overlapping pre- and post-expansion images. I cannot find such quantitative measures in this paper. Or, instead, since the authors developed the image processing software, the distributions of SYCP3 strand distances can be used to quantitatively evaluate the isotropicies of proExM, U-ExM and MAP at the ultrastructural level. Since the expansion factors differ for proExM, U-ExM and MAP, I suggest to normalize each distribution by the corresponding expansion factor for comparison of the three expansion protocols.

We agree with the author that it is necessary to compare and analyze correlated pre- and post-expansion images to testify the isotropy of expansion protocols. We therefore performed a new experiment that analyzes expansion-related distortions in RCM images of the same cells imaged pre- and post-expansion. The transformation matrices show low distortions introduced by the physical expansion process (see new Supplementary Figs. 6 and 7)

4. From a single profile example in Fig. 4d, the authors claimed “far more complex distribution of the central region proteins than the bimodal distribution”. Simply saying “more complex” is not helpful to figure out the differences. Describe more specifically how the distribution differ from bimodal. Also, prove the higher complexity with more data. One example image is not enough.

We recently published an EM study on the quantitative basis of meiotic chromosome synapsis in Scientific Reports (Spindler et al., 2019). In this study we generated EM tomography based 3D models of the SC from which we derived quantitative and topological data. Hereby, we specifically analyzed the data for the previously suggested bilayered organization of the transverse filaments in mouse. In brief, the individual morphological components of the SC (lateral elements, central element, transverse filaments) were segmented manually based on continuity in 3D. The command line program ‘model2point’ was used to generate a point representation of the segmented structures. Under the assumption that the transverse filaments are organized into two layers in the SC of mouse, we fitted two planes through the endpoints of the transverse filaments (treating the two sets of filaments on either side of the central element independently). The planes were fitted to minimize the squared orthogonal distance of each point to the plane. The proposed bimodal distribution would be reflected in two approximately parallel planes. Our data however did not follow a bimodal distribution. Instead the fit supports the organization of the TFs into a single layer for all organized tomograms. The resolution power of EM tomography allowed us to complement the robust mathematical analysis with a visual analysis of the individual transverse filaments. In accordance with the mathematical analysis, no distinct layers were apparent. The individual filaments are distributed rather randomly in a far more complex distribution than a bilayer.

The analysis of lateral view sections of the expanded SC supports these EM findings. Overall, the N-terminus of the transverse filament protein SYCP1 shows a monomodal signal distribution in lateral views in accordance with the single layer of transverse filaments revealed by EM tomography. Occasionally, the high labeling density provided by MAP combined with the post-processing super-resolution of SIM enables even the resolution of individual transverse filaments which is reflected in respective bi- and multimodal signal distributions. We have added a short paragraph introducing our

EM tomography derived findings in more depth. We have further related these findings to the results of the intensity profiles of the N-terminus of the transverse filament protein SYCP1 on expanded SCs, now highlighting that the averaged intensity profiles of the SYCP1 N-terminus show a monomodal distribution, while individual intensity profiles reflect a bi- or even multimodal distribution due to the resolution of individual filaments by expansion microscopy.

Furthermore, we analyzed more data highlighting the more complex distribution of the central region proteins than the bimodal distribution (see new Supplementary Fig. 10). We revised the text accordingly. It reads now:

“Therefore, we next analyzed the signal distribution of the N-terminus of the transverse filament protein SYCP1 by MAP-SIM of nuclear spreadings in more detail. In addition to analyzing the distribution of proteins in frontal views, we also generated line profiles of SC proteins from lateral views (Figs. 4a-c, Supplementary Fig. 10, and Supplementary Movie 1). In agreement with EM tomographic data³⁸, the signal distributions of the MAP-SIM imaged SYCP1 N-terminus and SYCE3 indicated a far more complex distribution of the central region proteins than the bimodal distribution previously described by dSTORM and immunogold EM^{19,20}. In some lateral view sections of MAP-SIM images of the SYCP3 signal the N-terminus of the transverse filament protein SYCP1 and the central element protein SYCE3 revealed bi- and multimodal signal distributions supporting a multilayered organization of the central element (Fig. 4d and Supplementary Fig. 10). On the other hand, averaged cross-sectional intensity profiles of SYCE3 and SYCP1N (Figs. 4e,f) clearly showed a monomodal distribution of the two proteins (Fig. 4g and Supplementary Fig. 10) in accordance with the unstructured organization of the transverse filaments into a large single layer as revealed by EM tomography in a previous study²⁷. The occasional disclosure of multimodal distributions in some lateral view sections of the SC in MAP-SIM views can be attributed to the high labeling density provided by MAP. Combined with the post-processing super-resolution of SIM, MAP-SIM enables even the resolution of individual transverse filaments which is reflected in the few detected multimodal distributions.”

5. The only obviously new structures in the paper is the bifurcation and fraying of the two SC strands in Fig. 5. I am not an expert in meiosis to judge the importance of the findings. The authors discussion on these structures sounds descriptive to me. What would be the biological, functional implications of these new findings?

We thank the reviewer for his/her remark. The bifurcation and fraying of the SC strands has previously only been observed on the EM level (Dresser et al., 1980; Nebel et al., 1962; Del Mazo et al., 1986). Here, a schematic model shows the possible organization of the lateral elements based on the observed splitting. Each of the lateral elements in the model is divided into two sub lateral elements which are each associated with a sister chromatid (del Mazo et al., 1987). It has been speculated that the proteinous boundary between the sister chromatids of a single homolog could prevent recombination between the sister chromatids (Syrjänen et al., 2014). The fraying could also reflect the onset of the disassembly of the axes. However, evidence for these hypothesis remains missing. Expansion microscopy, for the first time, resolves the bifurcation and fraying of the SYCP3 positive SC strands on the light microscopic level. Additional experiments would be required to challenge the hypothesis of intra-homolog recombination prevention through splitting of the SC strands. As a first step, the topological relation of the telomeric repeats with the frayed ends of the SYCP3 positive strands could be analyzed by performing expansion fluorescence in situ hybridization (ExFISH) in combination with localizing SYCP3. However, the aim of this study was not directed at uncovering the reasons for the splitting of the SC strands but to show that these rare nanoscale features whose resolution was previously reserved for EM techniques can be resolved by expansion microscopy as a starting point for further investigations. Moreover, the splitting of the SC strands is an example for the degree of molecular detail that can be resolved with MAP-SIM to uncover new

areas of investigation in cell biology. We have included a paragraph to the text in the revised version of our manuscript to indicate the potential of resolving such high molecular detail to diverse biological questions.

Followings are minor comments.

a) The following sentence in the introduction does not make sense. "The spreading of the SCs directly onto the surface of the coverslip results in the localization of the proteins close to the coverslip post-expansion and allows the use of a water-immersion objective for imaging of hydrogels without spherical aberrations."

If the sample is placed close to the surface, even oil immersion lens can be used. Did the authors mean that high NA lenses can be used?

We agree and rephrased the sentence. It reads now: "The spreading of the SCs directly onto the surface of the coverslip results in the localization of the proteins close to the coverslip post-expansion and thus avoids dehydration and cryosectioning of the sample enabling the use of high numerical aperture objectives for imaging of hydrogels without spherical aberrations."

b) In the last paragraph of the introduction, the authors claim the resolution of ExSTORM is 20-30 nm, which is typical resolution for unexpanded STORM. The ExSTORM paper claims for 10-20 nm resolution. Do the authors of this paper have evidences supporting that ExSTORM resolution is worse than reported?

The reviewer is right; the ExSTORM paper determined the lateral resolution from single-color experiments with SYCP3 to ~ 15 nm. We therefore corrected the sentence in the introduction.

c) High-resolution STORM may be limited in two colors. But, three-color DNA-PAINT is possible with exchangePAINT and offers even higher resolution than STORM. What would be the advantages of 3-color MAP-SIM over 3-color DNA-PAINT?

DNA-PAINT is a very powerful single-molecule localization microscopy method which achieves currently the highest spatial resolution (ignoring MINIFLUX) but it also requires labeling of antibodies with oligonucleotides while MAP-SIM works with commercial available antibodies. The major problem with using DNA-PAINT on expanded samples is, however, that the structures of interest are usually not in direct contact to the coverslip but located at a distance from it. Hence, TIRF illumination cannot be used, which is a prerequisite to perform DNA-PAINT with fluorescently labeled oligonucleotides. MAP-SIM provides 3D 3-color images of samples located at a distance from the coverslip limited only by the working distance of the water-immersion objective without spherical aberrations.

d) Online Methods, "Immunofluorescence of non-expanded SCs": What is NGS? Normal goat serum?

We thank the reviewer and added the information. NGS is normal goat serum.

e) Figure 3. Caption: There is no "g" panel in the figure. Perhaps, the "g" in the caption is typo for "f".

We thank the reviewer for careful reading and corrected the Figure Caption, i.e. we exchanged "g" by "f".

References

- Hernández-Hernández, A., Masich, S., Fukuda, T., Kouznetsova, A., Sandin, S., Daneholt, B., & Höög, C. (2016). The central element of the synaptonemal complex in mice is organized as a bilayered junction structure. *J Cell Sci*, *129*(11), 2239-2249.
- Schücker, K., Holm, T., Franke, C., Sauer, M., & Benavente, R. (2015). Elucidation of synaptonemal complex organization by super-resolution imaging with isotropic resolution. *Proceedings of the National Academy of Sciences*, *112*(7), 2029-2033.
- Harris, L. J., Skaletsky, E., & McPherson, A. (1998). Crystallographic structure of an intact IgG1 monoclonal antibody. *Journal of molecular biology*, *275*(5), 861-872.
- Locker, J. K., & Schmid, S. L. (2013). Integrated electron microscopy: super-duper resolution. *PLoS biology*, *11*(8), e1001639.
- Roux, K. H. (1999). Immunoglobulin structure and function as revealed by electron microscopy. *International archives of allergy and immunology*, *120*(2), 85-99.
- Wolf, E., Kastner, B., & Lührmann, R. (2012). Antisense-targeted immuno-EM localization of the pre-mRNA path in the spliceosomal C complex. *rna*, *18*(7), 1347-1357.
- Spindler, M. C., Filbeck, S., Stigloher, C., & Benavente, R. (2019). Quantitative basis of meiotic chromosome synapsis analyzed by electron tomography. *Scientific reports*, *9*.
- Dresser, M. E., & Moses, M. J. (1980). Synaptonemal complex karyotyping in spermatocytes of the Chinese hamster (*Cricetulus griseus*). *Chromosoma*, *76*(1), 1-22.
- Nebel, B. R., & Coulon, E. M. (1962). The fine structure of chromosomes in pigeon spermatocytes. *Chromosoma*, *13*(3), 272-291.
- Del Mazo, J., & Gil-Alberdi, L. G. (1986). Multistranded organization of the lateral elements of the synaptonemal complex in the rat and mouse. *Cytogenetic and Genome Research*, *41*(4), 219-224.
- Syrjänen, J. L., Pellegrini, L., & Davies, O. R. (2014). A molecular model for the role of SYCP3 in meiotic chromosome organisation. *Elife*, *3*, e02963.

REVIEWERS' COMMENTS:

Reviewer #1 (Remarks to the Author):

The changes and additions made to the manuscript in this revision have clarified many points and strengthened the evidence in places where it was insufficient, and all of the points I raised have been answered to my satisfaction. —PC

Reviewer #2 (Remarks to the Author):

The authors have sufficiently addressed my concerns. I have attached one comment regarding an response that could be addressed in the text.

Attached below:

The authors state that “post-expansion labeling protocols substantially increases the epitope accessibility for IgG antibodies resulting in higher labeling densities especially of sterically demanding multiprotein complexes” This is a significant claim and perhaps one of the most important takeaways from the paper, as super-resolution imaging is fundamentally limited by labeling density. Can the authors provide quantitation to this point? Specifically, does ExM-SIM offer higher labeling densities than STORM in 1 color (STORM is pre-labeled). If so, can higher resolutions be achieved, or are there more continuous structures. Can a labeling density metric be derived?

We fully agree with the reviewer that our finding of the advantage of post-expansion labeling regarding labeling efficiency especially of sterically demanding multiprotein complexes is one of the most important results of our manuscript. The comparison of labeling density of unexpanded versus expanded samples is, however, a very complex endeavor. First, the antibody binding constant can differ after protein denaturation as used by MAP. Hence, the binding efficiency of the same antibody can vary before and after expansion. Second, SIM is per se not single-molecule sensitive as (d)STORM is thus rendering the direct comparison difficult. In fact, only the labeling continuity and signal density along a filamentous structure like the SC can be compared, and this is what we did to compare the different expansion protocols. Due to this reasons the advantage of post-expansion labeling might be less noticeable in most experiments. Only in sterically demanding very protein dense regions of multiprotein complexes the epitope accessibility might be improved upon expansion, presupposed that the epitopes survive denaturation and expansion and are still recognized by the antibody. In the case of the SC the multilayered organization of proteins in the central element and SYCP3 fraying of the axes represent such cases where post-expansion labeling outperforms in some regions pre-expansion labeling. To cut a long story short, we fully agree with the reviewer and acknowledge his/her ideas to further increase the impact of our manuscript but the question is presently too complex to be answered correctly. The other question whether post-expansion labeling can achieve a higher resolution has to be answered with yes mainly due to the fact that the linkage error (the distance between the fluorophore and the protein of interest) decreases substantially when using post-expansion labeling. If as in our studies, proteins are labeled by immunolabeling pre-expansion the linkage error is $\sim 20\text{-}30$ nm (primary and secondary IgG antibody with a size of $10\text{-}15$ nm). This linkage error will “expand” during expansion – in our case ~ 4 fold to $80\text{-}120$ nm whereas post-expansion labeling will reduce the ‘true’ linkage error to $\sim 5\text{-}8$ nm for a 4-fold expanded sample. We highlighted these findings in more detail in the revised version of our manuscript.

I accept the authors claim that this may be beyond the scope of this manuscript. However, could the authors elaborate on this in the discussion here:

“Our results demonstrate that post-expansion labeling protocols reduce the linkage error and substantially increase the epitope accessibility for IgG antibodies resulting in higher labeling densities especially of sterically demanding multiprotein complexes^{14,29}”

It's probably difficult to claim substantially increased epitope accessibility without actually making the comparison. I think a less strong claim (i.e. more continuous labeling suggests higher accessibilities, although this is a challenging expt and requires future measurements in different epitope contexts).

Reviewer #3 (Remarks to the Author):

The revised version of the manuscript carefully addresses the issues that I raised previously. Especially, the new fitting schemes for the distance histograms for quantifying the strand distance is well validated with simulation of a predefined helical structure and resulted in consistent results for both 1- and 2-color images. Also, the pre- and post-expansion alignment is properly executed and highly informative.

However, I found a few points in the revised manuscript that have to be corrected or addressed.

(1) The following paragraph added to the revised manuscript contain inconsistent figure call-outs. "Fig. 4d" should be corrected to "Fig. 4g". "Fig. 4g and Supplementary Fig. 10" should be corrected to "Fig. 4d".

In some lateral view sections of MAP-SIM images of the SYCP3 signal the N-terminus of the transverse filament protein SYCP1 and the central element protein SYCE3 revealed bi- and multimodal signal distributions supporting a multilayered organization of the central element (Fig. 4d and Supplementary Fig. 10). On the other hand, averaged cross-sectional intensity profiles of SYCE3 and SYCP1N (Figs. 4e,f) clearly showed a monomodal distribution of the two proteins (Fig. 4g and Supplementary Fig. 10) in accordance with the unstructured organization of the transverse filaments into a large single layer as revealed by EM tomography in a previous study²⁷.

(2) The author's reply to my major comment #5 is largely satisfactory. However, I cannot find the "new paragraph" that the authors mentioned in the last sentence of their reply. I just found one new sentence in the revised paragraph. Where is the "new paragraph"? Since the journal does not limit the length of the texts, can the authors consider adding some of the rebuttal comments in the manuscript to help readers unfamiliar to the biological system like myself?

(3) The author's comment to my minor comment (c) about DNA-PAINT's requirement for TIRF is correct. However, TIRF is compatible with their particular application of imaging SCs spread out on the coverslip. Therefore, for many systems to which TIRF is applicable, including SCs, I don't see the advantage of MAP-SIM over Exchange-PAINT.

In addition, MAP-SIM depth is shorter than regular SIM. Theoretically, MAP-SIM depth is the SIM depth divided by the expansion factor. For instance, the working distance of a high-NA objective for SIM is ~200 micron. Regarding an expansion factor of 4, the depth limit would be ~50 micron. Moreover, the spherical aberration of the expanded gel may be different from unexpanded samples in aqueous media. It is true that even after considering expansion and aberration, the imaging depth of MAP-SIM is better than DNA-PAINT. It is noteworthy to mention these for guiding readers considering MAP-SIM for their systems of interest.

Point-by-point response to the referees' comments:

We wish to thank the reviewers for their positive and constructive comments that have helped to improve the manuscript and strengthen its novelty, performances and the fields of applications. We have revised the manuscript in depth taking into account all the remarks.

Reviewer #1 (Remarks to the Author):

The changes and additions made to the manuscript in this revision have clarified many points and strengthened the evidence in places where it was insufficient, and all of the points I raised have been answered to my satisfaction. —PC

We thank Peter Carlton for his comments that helped to improve the quality of our manuscript.

Reviewer #2 (Remarks to the Author):

The authors have sufficiently addressed my concerns. I have attached one comment regarding a response that could be addressed in the text.

It's probably difficult to claim substantially increased epitope accessibility without actually making the comparison. I think a less strong claim (i.e. more continuous labeling suggests higher accessibilities, although this is a challenging expt and requires future measurements in different epitope contexts).

We fully agree with the reviewer and changed the wording accordingly in our manuscript.

“Our results demonstrate that post-expansion labeling protocols reduce the linkage error. Furthermore, the more continuous labeling of SC filaments observed by post-expansion labeling suggests a higher epitope accessibility for IgG antibodies resulting in improved labeling densities especially of sterically demanding multiprotein complexes^{14,29}. Consequently, MAP-SIM provides enhanced ultrastructural resolution as demonstrated here for an important DNA-associated multiprotein complex.

Reviewer #3 (Remarks to the Author):

The revised version of the manuscript carefully addresses the issues that I raised previously. Especially, the new fitting schemes for the distance histograms for quantifying the strand distance is well validated with simulation of a predefined helical structure and resulted in consistent results for both 1- and 2-color images. Also, the pre- and post-expansion alignment is properly executed and highly informative. However, I found a few points in the revised manuscript that have to be corrected or addressed.

(1) The following paragraph added to the revised manuscript contain inconsistent figure call-outs. “Fig. 4d” should be corrected to “Fig. 4g”. “Fig. 4g and Supplementary Fig. 10” should be corrected to “Fig. 4d”.

In some lateral view sections of MAP-SIM images of the SYCP3 signal the N-terminus of the transverse filament protein SYCP1 and the central element protein SYCE3 revealed bi- and multimodal signal distributions supporting a multilayered organization of the central element (Fig. 4d

and Supplementary Fig. 10). On the other hand, averaged cross-sectional intensity profiles of SYCE3 and SYCP1N (Figs. 4e,f) clearly showed a monomodal distribution of the two proteins (Fig. 4g and Supplementary Fig. 10) in accordance with the unstructured organization of the transverse filaments into a large single layer as revealed by EM tomography in a previous study²⁷.

We thank the reviewer for careful reading and corrected the manuscript accordingly.

(2) The author's reply to my major comment #5 is largely satisfactory. However, I cannot find the "new paragraph" that the authors mentioned in the last sentence of their reply. I just found one new sentence in the revised paragraph. Where is the "new paragraph"? Since the journal does not limit the length of the texts, can the authors consider adding some of the rebuttal comments in the manuscript to help readers unfamiliar to the biological system like myself?

The reviewer is right, we did indeed not add a new paragraph. But we checked again our revised version and are convinced that it contains all information even for a less biological experienced reader. The message of the manuscript at this point is that MAP-SIM can uncover new details of the molecular organization of SCs. The resolution of the splitting of the SC strands is an example for the degree of molecular detail that can be resolved using MAP-SIM overall to uncover new areas of investigation in cell biology.

(3) The author's comment to my minor comment (c) about DNA-PAINT's requirement for TIRF is correct. However, TIRF is compatible with their particular application of imaging SCs spread out on the coverslip. Therefore, for many systems to which TIRF is applicable, including SCs, I don't see the advantage of MAP-SIM over Exchange-PAINT.

In addition, MAP-SIM depth is shorter than regular SIM. Theoretically, MAP-SIM depth is the SIM depth divided by the expansion factor. For instance, the working distance of a high-NA objective for SIM is ~200 micron. Regarding an expansion factor of 4, the depth limit would be ~50 micron. Moreover, the spherical aberration of the expanded gel may be different from unexpanded samples in aqueous media. It is true that even after considering expansion and aberration, the imaging depth of MAP-SIM is better than DNA-PAINT. It is noteworthy to mention these for guiding readers considering MAP-SIM for their systems of interest.

One of the important advantages of MAP-SIM is that post-expansion labeling enables higher labeling densities and thus enhanced ultrastructural resolution as demonstrated for a few molecular details of SC organization in our manuscript. Exchange-PAINT achieves a similar resolution as dSTORM as long as multiple-oligo-labeled antibodies are used for immunolabeling because in multiple labeled antibodies it is the size of the antibody which determines the achievable resolution and not the localization precision of an individual dye and oligo, respectively. We believe that this is an important point that has to be considered in the argumentation when comparing dSTORM with DNA-PAINT. But of course, the reviewer is right, imaging depth is finally restricted by the working distance of the objective, which is in our case 280 μm and aberration corrected. That is, for a 4.2x expanded sample we can image up to depths of $280 \mu\text{m}/4.2 = 66.7 \mu\text{m}$.

An axial depth of ~67 μm is large enough to enable imaging of the SCs of a whole set of chromosomes in a spermatocyte with detailed structural information of single chromosomes.

"Intriguingly, spreading of SCs in combination with MAP allowed us to acquire 3D super-resolution images of large expanded samples, e.g. $100 \times 100 \times 30 \mu\text{m}^3$ (200 nm z-steps) by re-scan confocal microscopy (RCM)³⁵ (Figs. 3d,e) and $80 \times 80 \times 15 \mu\text{m}^3$ (110 nm z-steps) and larger by SIM, i.e. imaging of the SCs of a whole set of chromosomes in a spermatocyte with detailed structural information of single chromosomes (Fig. 3f)."

Nevertheless, we added a sentence to the Conclusions to highlight that the axial depth of imaging is limited dependent on the used microscope objective.

“We have shown that through isotropic structure preserving expansion combined with SIM, ultrastructural details of SCs can be revealed with standard immunofluorescence techniques in common sample preparations at axial depths of up to $\sim 60 \mu\text{m}$ using high numerical aperture water immersion microscope objectives.”